# MATCHFIXAGENT: Language-Agnostic Autonomous Repository-Level Code Translation Validation and Repair

**Ali Reza Ibrahimzada** [1] [†]  **Brandon Paulsen** [2]  **Reyhaneh Jabbarvand** [1]  **Joey Dodds** [2]  **Daniel Kroening** [2] [3]

## Abstract

Code translation transforms source code from one programming language (PL) to another. Validating the functional equivalence of translation and repairing, if necessary, are critical steps in code translation. Existing automated validation and repair approaches struggle to generalize to many PLs due to high engineering overhead, and they rely on existing and often inadequate test suites, which results in false claims of equivalence and ineffective translation repair. To bridge this gap, we develop MATCHFIXAGENT, a large language model (LLM)-based, PL-agnostic framework for equivalence validation and repair of translations. MATCHFIXAGENT features a multi-agent architecture that divides equivalence validation into several sub-tasks to ensure thorough and consistent semantic analysis of the translation. We compare MATCHFIXAGENT's validation and repair results with four repository-level code translation techniques. Our results demonstrate that MATCHFIXAGENT produces (in)equivalence verdicts for 99.2% of translation pairs, with the same equivalence validation result as prior work on 72.8% of them. When MATCHFIXAGENT's result disagrees with prior work, we find that 60.7% of the time MATCHFIXAGENT's result is actually correct. In addition, we show that MATCHFIXAGENT can repair 50.6% of inequivalent translation, compared to prior work's 18.5%.

## 1. Introduction

Code translation, the process of converting source code from one programming language (PL) to another, is a cornerstone of software modernization efforts that enhance performance, maintainability, and reliability (Khan et al., 2022; Jain & Chana, 2015; Jamshidi et al., 2013; Khadka et al., 2014). Translation validation and repair are integral steps in code translation for determining functional equivalence and patch generation for incorrect translations. However, performing validation and repair manually—particularly in large codebases—can be tedious, time-consuming, and error-prone, especially when complex code structures and dependencies are involved (Hora et al., 2015; Kula et al., 2018; Wang et al., 2020). Prior work defines value and type equivalences and translations to compare source and target implementations over pairs of concrete inputs. The inputs either come from existing tests from the source project (Shetty et al., 2024; Zhang et al., 2025; Wang et al., 2025a; Ibrahimzada et al., 2025) or differential fuzzing (Yang et al., 2025; Eniser et al., 2024). Despite notable advancements in validation and repair of repository-level code translation, existing techniques are hampered by the following limitations (examples given in Appendix B):

1. *Difficulty Generalizing to Many PL Pairs.* While the fundamental ideas of current validation approaches extend to many PL pairs, their actual implementations typically support just one language pair. This is because supporting language interoperability between a pair of PLs requires a large engineering effort, as evidenced by the size of these tools[1]. Given the quadratic number of PL pairs, language interoperability techniques are extremely challenging to scale to many PL pairs.

2. *Unknown Test Requirements.* Current translation validation approaches require a set of valid inputs to validate the input-output equivalence between source functions and their translation. They generate these inputs by either executing available source tests (Wang et al., 2025a; Zhang et al., 2025; Ibrahimzada et al., 2025; Shetty et al., 2024) or fuzzing the source project (Eniser et al., 2024; Yang et al., 2025) Unit tests are often incomplete, missing important inputs, and resulting in false claims of equivalence. Fuzzing techniques suffer from generating invalid inputs, also resulting in false claims of inequiva-

---

[†]Work done when author was an intern at AWS. [1]Siebel School of Computing and Data Science, University of Illinois Urbana-Champaign, Urbana, IL, USA [2]Amazon, Arlington, VA, USA [3]University of Oxford, Oxford, UK. Correspondence to: Ali Reza Ibrahimzada <alirezai@illinois.edu>.

*Proceedings of the 43rd International Conference on Machine Learning*, Seoul, South Korea. PMLR 306, 2026. Copyright 2026 by the author(s).

---

[1]The implementation of notable recent translation and validation techniques are ALPHATRANS (10859 LoC), OXIDIZER (19052 LoC), and SKEL (3843 LoC).

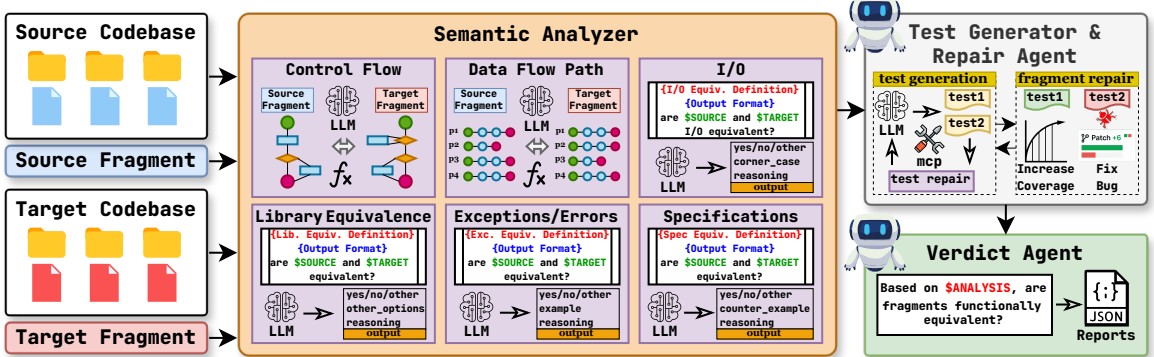

*Figure 1.* Overview of MATCHFIXAGENT.

lence, and in general, failing to reach deep into the code or create complex objects in the context of real-world projects (Klees et al., 2018; Godefroid et al., 2008).

3. *Ineffective Translation Repair.* Recent studies have shown that a more rigorous validation can reveal more translation bugs (Pan et al., 2024). Hence, advancement in translation validation should be accompanied by effective repair strategies. Existing techniques, however, either require the user to fix incorrect translations manually (Wang et al., 2025a) or use feedback-driven re-prompting strategies (Zhang et al., 2025; Ibrahimzada et al., 2025) that are barely effective in repository-level code translation due to long call-chain dependencies in real-world projects (Ibrahimzada et al., 2025).

Prior work consists of high-effort implementations that can deliver inconsistent results and only apply to one of a quadratic number of language pairs. Large Language Models (LLMs) have recently been successful at same-language equivalence validation (Wei et al., 2025a; Maveli et al., 2025), so replacing cross-language equivalence implementations with LLM decisions is a logical next step. While there is a risk of incorrect results, the baseline mechanical approaches already display low accuracy.

Towards this end, we propose MATCHFIXAGENT, a *language-agnostic* approach to automate *validation* and *repair* of repository-level code translation (§2.1). MATCHFIXAGENT combines the generative power of LLMs with several approximate code semantic analyses, i.e., analysis of control- and data-flow paths, library APIs, exception handling, and (formal or informal) specification (§2.2). These semantic analyses are then fed to a *test generator & repair agent* to generate tests for assessing functional equivalence, and repair the translation in the case of failing tests (§2.3). The final equivalence decision will be made by the *verdict* agent, considering the approximate semantic analyses and test execution results (§2.4). MATCHFIXAGENT is very lightweight (1,650 lines of code), modular, and interoperable with existing repository-level translation systems.

We evaluate the effectiveness of MATCHFIXAGENT for repository-level translation validation and repair against four existing techniques (Ibrahimzada et al., 2025; Zhang et al., 2025; Wang et al., 2025a; Ou et al., 2025). Our benchmark comprises 2,219 source–translation function pairs, which cover 6 PL pairs, drawn from 24 real-world projects totaling over $900K$ lines of code (details in Appendix C). For each translation pair, we obtain an equivalence verdict (validation outcome) from both MATCHFIXAGENT and other techniques. Overall, MATCHFIXAGENT returns a verdict for 99.2% of pairs, while alternative approaches do so for only 71.6% (§3.1). On the 1,571 pairs where both produce verdicts, MATCHFIXAGENT agrees with other approaches in 72.8% of cases. For the remaining disagreements, a systematic manual investigation finds MATCHFIXAGENT to be correct in 60.7% of cases and incorrect in the rest (§D.1). In translation repair, MATCHFIXAGENT can fix 50.6% of translation bugs, 32.1% more than existing approaches (§3.2). We show that MATCHFIXAGENT is compatible with different LLMs and agent frameworks, producing comparable results (§3.3). Lastly, our ablation study shows that removing code analyses and in-the-loop test generation reduces verdict accuracy by 42.3%, while increasing token usage by 5.2% (§3.4). These results confirm that MATCHFIXAGENT is a viable alternative to prior work's validation and repair approaches, while being vastly easier to adapt to new PL pairs. Our notable contributions are:

1. We present MATCHFIXAGENT, a PL-agnostic, agentic approach for validation and repair of repository-level code translation.

2. We demonstrate that MATCHFIXAGENT is a viable alternative to prior work's validation approach, while being vastly easier to adapt to new PL pairs.

3. We show the benefit of MATCHFIXAGENT's multi-agent architecture compared to standalone-agent design.

## 2. MATCHFIXAGENT

### 2.1. Overview

Figure 1 gives an overview of MATCHFIXAGENT, which consists of three main components: (1) the Semantic Analyzer (§2.2), (2) the Test Generator & Repair Agent (§2.3),

**Algorithm 1** MATCHFIXAGENT

```
 1: Input: sourceProject, sourceFunc, translatedProject,
 2:        translatedFunc, LLM, tools, timeout
 3: Output: validationRepairReport
 4: transPair ← [sourceFunc, translatedFunc]
 5: function SEMANALYZER(transPair, LLM):
 6:    cfgSrc, dfSrc ← build_cfg(sourceFunc)
 7:    cfgTgt, dfTgt ← build_cfg(translatedFunc)
 8:    return {
 9:       controlFlowAnalyzer(cfgSrc, cfgTgt, transPair, LLM),
10:       dataFlowPathAnalyzer(dfSrc, dfTgt, transPair, LLM),
11:       ioAnalyzer(transPair, LLM),
12:       libraryAnalyzer(transPair, LLM),
13:       exceptionAnalyzer(transPair, LLM),
14:       specAnalyzer(transPair, LLM)}
15: end function
16: await semAnalysis ← SEMANALYZER(cfgSrc, dfSrc, cfgTgt, dfTgt)
17: testRepair ← testGenRepairAgent(prompt, LLM, tools, timeout)
18: verdict ← verdictAgent(semAnalysis, testRepair, LLM)
19: validationRepairReport ← semAnalysis ∪ testRepair ∪ verdict
20: return validationRepairReport
```

and (3) the Verdict Agent (§2.4). MATCHFIXAGENT takes as input a translation pair (source function and its translation) along with both projects, and outputs an equivalence verdict, a natural language report, and an optional repair patch. Algorithm 1 details the procedure.

The Semantic Analyzer invokes an LLM to analyze six semantic properties of the translation pair in parallel: control flow (§2.2.1), data flow (§2.2.2), input-output mapping (§2.2.3), library API usage (§2.2.4), exception handling (§2.2.5), and specifications (§2.2.6). This decomposition keeps the LLM focused and improves reliability (Anthropic, 2026a). Each sub-task outputs a report summarizing semantic differences. These reports are fed to the Test Generator & Repair Agent, which uses an off-the-shelf LLM coding agent (e.g., Claude Code (Anthropic, 2026c) or Codex (OpenAI, 2026a)) to write and execute tests that may reveal inequivalent behavior. If inequivalence is found, the agent attempts to repair the translation. Finally, the Verdict Agent validates the claims from the previous components and produces a final equivalence verdict with a summary.

### 2.2. Semantic Analyzer

The Semantic Analyzer takes as input the translation pair and an LLM. It first computes a control flow graph (CFG) and data flow graph (DFG) (described in §2.2.1 and §2.2.2 respectively), and then calls six sub-analyzers in parallel, each of which analyzes a different semantic property of the translation pair. Each sub-analyzer invokes the LLM with a custom prompt [2] describing the analysis to perform. The prompts are relatively simple and short. The prompt first defines a role for the LLM ("`You are an expert in...`"), a general definition of functional equivalence, and a specific definition of equivalence for the semantic property. It then instructs the analyzer to output an equivalence verdict and explanation for the specific semantic property it

---

[2]Please refer to our artifacts repository (Intelligent-CAT-Lab, 2026) for prompt structure of each semantic analyzer.

analyzed. In addition, certain analyzers output examples to demonstrate inequivalence. The final output of the Semantic Analyzer is a 6-tuple containing a JSON-formatted output of each sub-analyzer. The following subsections provide more details on the prompts of the six sub-analyzers.

#### 2.2.1. CONTROL FLOW ANALYZER

The *Control Flow Analyzer* is prompted to analyze the control flow structures of the source and translation to determine whether they are equivalent, looking for inequivalences such as reordered conditions, missing branches, or altered loop termination criteria. To aid this task, we provide the LLM with textual representations of the source and translation's CFGs. An example is shown in Figure 6. To compute the CFG, we use Tree-Sitter (Tree-Sitter, 2026) to construct an abstract syntax tree of the function, and then extract basic blocks and control flow structures. Tree-sitter supports 165+ PLs, making this process PL-agnostic. Each PL supported by MATCHFIXAGENT required approximately 280 lines of code, making it very easy to support many PLs.

To improve reliability and reduce costs, the control flow analyzer first (symbolically) computes a similarity score between the CFGs, and, if it falls above a threshold, it immediately returns an equivalent verdict without invoking the LLM. The overall procedure is shown in Algorithm 2. The analyzer abstracts each graph into canonical forms (lines 3–13) capturing node types (e.g., conditionals, loops, exception handlers) and edge types (control transfer relationships), then computes the structural similarity score based on the Jaccard index (Cha, 2007) (lines 14–16). We use 0.7 as the threshold. This results in approximately 25% of LLM invocations being skipped in our experiments, making this threshold relatively stringent.

#### 2.2.2. DATA FLOW ANALYZER

The *Data Flow Analyzer* is prompted to evaluate whether the flow of data within the source and translation is equivalent, looking for issues like unused variables. To aid the LLM, we provide textual representations of the source and translation's data flow graphs (DFGs). An example is shown in Figure 6. We keep our data flow computation extremely simple. For each statement in the AST, we extract variable names, label them as a def or a use, and associate them with a CFG node. We then extract def-use chains. This is primarily a syntactic analysis. We do not handle challenging problems such as aliasing, concurrency, or context sensitivity. The per-PL implementation effort is approximately 280 lines of code based on the six PLs supported by MATCHFIXAGENT. Similar to our control flow analyzer, we compute a similarity score between the DFGs, and short-circuit if it falls above a threshold. The analyzer, show in Algorithm 3, first extracts *def-use* chains for parameters and local vari-

**Algorithm 2** Control Flow Analyzer

1: **Input:** $cfgSource, cfgTarget, fragments, model, \tau{=}0.7$
2: **Output:** $cfgAnalysis$
3: **function** ABSTRACTGRAPH($cfg$):
4:    **for each** $(u, v)$ **with** $edge \in cfg$ **do**
5:      $uType, vType \leftarrow$ **classifyNode**($u$), **classifyNode**($v$)
6:      $edgeType \leftarrow$ **classifyEdge**($edge$)
7:      $nodes \leftarrow nodes \cup uType \cup vType$
8:      $edges \leftarrow edges \cup \langle uType, edgeType, vType \rangle$
9:    **end for**
10:    **return** $nodes, edges$
11: **end function**
12: $sourceNodes, sourceEdges \leftarrow$ ABSTRACTGRAPH($cfgSource$)
13: $targetNodes, targetEdges \leftarrow$ ABSTRACTGRAPH($cfgTarget$)
14: $nodeSim \leftarrow$ **jaccardSimilarity**($sourceNodes, targetNodes$)
15: $edgeSim \leftarrow$ **jaccardSimilarity**($sourceEdges, targetEdges$)
16: $similarity \leftarrow (0.5 \times nodeSim) + (0.5 \times edgeSim)$
17: **if** $similarity \geq \tau$ **then**
18:    $cfgAnalysis \leftarrow \{$"is_equivalent": "yes"$\}$
19: **else**
20:    $cfgAnalysis \leftarrow$ **LLM**($cfgSource, cfgTarget, fragments, model$)
21: **end if**
22: **return** $cfgAnalysis$

---

**Algorithm 3** Data Flow Path Analyzer

1: **Input:** $dfSource, dfTarget, fragments, model, \tau{=}0.7$
2: **Output:** $dfAnalysis$
3: **function** COMPUTEEDITDISTANCE($srcPath, tgtPath$):
4:    $sim\_a \leftarrow 0$
5:    **for each** $xPath \in srcPath$ **do**
6:      $best \leftarrow 0$
7:      **for each** $yPath \in tgtPath$ **do**
8:        $score \leftarrow$ **jaccardSimilarity**($xPath, yPath$)
9:        $best \leftarrow$ **max**($best, score$)
10:      **end for**
11:      $sim\_a \leftarrow sim\_a + best$
12:    **end for**
13:    $sim\_a \leftarrow sim\_a/|srcPath|$
14:    $sim\_b \leftarrow$ repeat loop with srcPath and tgtPath swapped
15:    **return** $(sim\_a + sim\_b)/2$
16: **end function**
17: $srcPath, tgtPath \leftarrow$ **getVariablePaths**($dfSource, dfTarget$)
18: $similarity \leftarrow$ COMPUTEEDITDISTANCE($srcPath, tgtPath$)
19: **if** $similarity \geq \tau$ **then**
20:    $dfAnalysis \leftarrow \{$"is_equivalent": "yes"$\}$
21: **else**
22:    $dfAnalysis \leftarrow$ **LLM**($dfSource, dfTarget, fragments, model$)
23: **end if**
24: **return** $dfAnalysis$

---

ables, capturing how data values are defined, propagated, and consumed. The extracted paths are compared using edit distance (Miller et al., 2009) as the similarity measure (lines 3–18). We again use 0.7 as the threshold, which results in approximately 35% of LLM invocations being skipped.

### 2.2.3. IO ANALYSIS

The *IO Analyzer* is prompted to evaluate whether the observable input-output behavior of the source and target fragments is semantically equivalent. The prompt includes an IO equivalence definition, which assess five dimensions: (1) semantic equivalence of accepted inputs, (2) consistency of produced outputs, (3) preservation of side effects (e.g., file operations, network calls, or global state modifications), (4) uniform handling of edge cases, and (5) similarity in performance-critical complexity. The LLM is prompted also prompted to produce a plausible input that would trigger dissimilar IO behavior if it believes the translation is in-

equivalent. This methodology catches inequivalences such as differing error messages, inconsistent encoding assumptions, or missing side effects—often overlooked by structural analyses (§2.2.1, §2.2.2) alone.

### 2.2.4. LIBRARY ANALYZER

The *Library API Analyzer* is prompted to consider the behavior of external library APIs called in the source and translation, and evaluate whether their differences result in inequivalent behavior. This analyzer primarily detects subtle differences between similar library APIs in the source and translation. It provides suggestions to fix inequivalent behavior as well.

### 2.2.5. EXCEPTION & ERROR ANALYZER

The *Exception and Error Handling Analyzer* is prompted to validate whether error detection, exception raising, and error recovery mechanisms in the source and target code fragments are functionally equivalent. The prompt include five dimensions for equivalence: (1) detecting and handling the same error conditions, (2) using semantically equivalent exception/error types, (3) producing equivalent error messages or codes, (4) preserving consistent recovery mechanisms, and (5) propagating errors in equivalent ways. If neither fragment implements explicit error handling, the analyzer deems them equivalent for this dimension. Otherwise, it statically identifies exception constructs (e.g., `try-catch`, `throw`, `return`-error patterns) and uses LLM reasoning to compare semantics. For instance, if the source raises a specific `FileNotFoundError` while the target raises a generic `IOException`, the discrepancy is flagged, as it may affect upstream handling. Where differences exist, the analyzer also recommends target-language error handling constructs that align with the source's semantics.

### 2.2.6. SPECIFICATIONS ANALYZER

The *Specification Analyzer* is prompted to assess whether the source and target code fragments adhere to the same explicit or implicit functional specifications. The prompt includes Specification equivalence definition which state that the source and translation should: (1) fulfill the same documented or inferred functional requirements, (2) satisfy identical pre-conditions and post-conditions, (3) maintain the same invariants, and (4) handle the same input domain, including edge cases. The LLM is instructed to extract available specifications from function signatures, type annotations, and relevant comments, or, when no formal documentation exists, infer behavioral contracts from code semantics. The LLM is asked to compare the contracts of the source and translation. For example, if the source specifies "`returns 1 on success, 0 on failure`" and the target returns `Boolean` values, the inconsistency is flagged. In such cases,

the LLM is instructed to produce a formalized specification that reconciles both implementations and provides counterexamples demonstrating behavior mismatch.

### 2.3. Test Generator & Repair Agent

The *Test Generator & Repair Agent* uses an off-the-shelf LLM coding agent and the reports from the Semantic Analyzer to write and execute tests that demonstrate functional (in)equivalence. This agent helps catch hallucinations and confirm the claims in the Semantic Analyzer reports. The prompt for the agent is shown in Figure 7, which includes a definition of equivalence, instructions to write tests in both the source and target PL that test the (in)equivalence of the translation, and finally instructions to repair the translation if it is not equivalent. We use Claude Code (Anthropic, 2026c) as the agent for most of our experiments, which comes with a set of tools out of the box, namely, reading + writing files, executing arbitrary shell commands, and searching the web. The agent outputs an overall equivalence verdict, a set of tests in both the source and target PL, and a translation patch if the agent believed the translation was not equivalent.

### 2.4. Verdict Agent

The final component of MATCHFIXAGENT is the Verdict Agent, which produces a definitive assessment of the translation's correctness by synthesizing the information from the previous two stages. The Verdict Agent takes as input the semantic analysis report and the test execution + repair report. It leverages another LLM agent to consolidate these results into a final verdict. This agent's primary job is to (1) confirm the results of the Test Generator & Repair Agent, and (2) to produce a condensed summary of the results, which is useful for end-users.

## 3. Evaluation

We conducted an extensive set of empirical studies, evaluating the effectiveness of MATCHFIXAGENT in translation validation (§3.1), translation repair (§3.2), investigation of development cost and adaptability (§3.3), and ablation studies to justify the architecture of MATCHFIXAGENT (§3.4). We evaluate MATCHFIXAGENT on benchmarks used in prior work (Ibrahimzada et al., 2025; Wang et al., 2025a; Zhang et al., 2025; Ou et al., 2025) and compare against them in translation validation and repair. For LLM and agentic framework, we used `Claude 3.7 Sonnet` (Anthropic, 2026b) and Claude Code (Anthropic, 2026c). More experimental setup details are listed in Appendix C.

### 3.1. Effectiveness in Translation Validation

To assess effectiveness in translation validation, we run MATCHFIXAGENT on each translation pair to obtain an equivalence verdict, and compare it with the verdict of existing tools. The columns under **Tool Validation** and **MATCHFIXAGENT** in Table 1 summarize the equivalence verdicts for the competing validation tool and MATCHFIXAGENT, respectively. There are three types of equivalence verdicts: (1) Equivalent (**EQ**) indicating the source function and its translation are equivalent, (2) Not Equivalent (**NEQ**) indicating they are inequivalent, and (3) Validation Failure (**VF**) indicating that the tool failed to provide a verdict. Competing tools can fail to provide a verdict if (1) the source project did not have unit tests covering the function, or (2) the competing tool's language interoperability mechanism crashes before providing verdict. MATCHFIXAGENT can fail to provide a verdict if the timeout limit is reached. The **Agreement** column shows the number and percentage where both MATCHFIXAGENT's and others verdicts agree or disagree. The **Disagreement** column shows the result of our human investigation on disagreements. The **Tool** sub-column shows the percentage of disagreements where the existing tool's verdict was correct, and **Ours** sub-column shows the same metric for MATCHFIXAGENT. On average, MATCHFIXAGENT takes 309 seconds to produce a verdict, with an average cost of $1.22 and a total cost of $2,710.45.

These results demonstrate effectiveness of MATCHFIXAGENT in providing an equivalence verdict: MATCHFIXAGENT provides verdicts for 99.7% of translation pairs from ALPHATRANS and SKEL, whereas these techniques report verdicts for only 56.6% and 87.5% of their studied translation pairs, respectively. In addition, MATCHFIXAGENT's verdicts show a high level of agreement with all prior work, ranging from 63.7% to 80.2%. Furthermore, we manually investigate disagreements (§D.1) and show examples of incorrect verdicts (§D.2).

### 3.2. Effectiveness in Translation Repair

We investigate the effectiveness of MATCHFIXAGENT in automated repair of translation bugs, and its ability to improve code coverage of existing projects. Table 2 shows the results of this research question. To fairly evaluate the effectiveness of existing tools and MATCHFIXAGENT in translation repair, we extract a subset of translations where both techniques generated a patch.

In total, $\frac{265}{2219}$ (11.9%) buggy translations were considered for our study. To validate patches, we used original project tests that previously failed on buggy translations, and only considered a patch correct when all failing tests passed. We did not use generated tests by MATCHFIXAGENT to avoid bias in our evaluation. Column *Tool Repaired* shows the number of buggy translations repaired by existing techniques. Only $\frac{49}{265}$ (18.5%) bugs have been repaired by prior techniques. Except for RUSTREPOTRANS (Ou et al., 2025), other code translation tools failed to generate cor-

*Table 1.* Effectiveness of MATCHFIXAGENT in translation validation compared to existing techniques. **EQ:** Equivalent, **NEQ:** Not Equivalent, **VF:** Validation Failure, **Agreement**: number and percentage of translation pairs where MATCHFIXAGENT's and the existing tool's verdicts agree, **Disagreement**: percentage of disagreements ruled in favor of **Tool** and MATCHFIXAGENT (**Ours**). **VF**s are excluded from **Agreement** and **Disagreement** calculations.

| Tool | Project | Total # Trans. Pairs | Tool Validation | | | MATCHFIXAGENT | | | Agreement | Disagreement | |
|---|---|---|---|---|---|---|---|---|---|---|---|
| | | | EQ | NEQ | VF | EQ | NEQ | VF | | Tool | Ours |
| OXIDIZER | checkdigit | 29 | 21 (72.4) | 8 (27.6) | 0 (0) | 22 (75.9) | 7 (24.1) | 0 (0) | 24 (82.8) | 0.0 | 100 |
| | go-edlib | 24 | 18 (75) | 6 (25) | 0 (0) | 16 (66.7) | 8 (33.3) | 0 (0) | 14 (58.3) | 11.1 | 88.9 |
| | histogram | 19 | 12 (63.2) | 7 (36.8) | 0 (0) | 11 (57.9) | 7 (36.8) | 1 (5.3) | 8 (44.4) | 20.0 | 80.0 |
| | nameparts | 15 | 9 (60) | 6 (40) | 0 (0) | 12 (80) | 3 (20) | 0 (0) | 12 (80) | 33.3 | 66.7 |
| | stats | 53 | 38 (71.7) | 14 (26.4) | 1 (1.9) | 37 (69.8) | 16 (30.2) | 0 (0) | 35 (67.3) | 0.0 | 100 |
| | textrank | 52 | 40 (76.9) | 12 (23.1) | 0 (0) | 34 (65.4) | 18 (34.6) | 0 (0) | 28 (53.8) | 42.9 | 57.1 |
| **Total** | | **192** | **138 (71.9)** | **53 (27.6)** | **1 (0.5)** | **132 (68.8)** | **59 (30.7)** | **1 (0.5)** | **121 (63.7)** | **15.9** | **84.1** |
| ALPHATRANS | cli | 273 | 210 (76.9) | 24 (8.8) | 39 (14.3) | 210 (76.9) | 60 (22) | 3 (1.1) | 176 (76.2) | 30.0 | 70.0 |
| | csv | 235 | 97 (41.3) | 61 (26) | 77 (32.8) | 185 (78.7) | 49 (20.9) | 1 (0.4) | 108 (68.8) | 30.0 | 70.0 |
| | fileupload | 192 | 19 (9.9) | 1 (0.5) | 172 (89.6) | 144 (75) | 48 (25) | 0 (0) | 16 (80) | 25.0 | 75.0 |
| | validator | 646 | 247 (38.2) | 103 (15.9) | 296 (45.8) | 483 (74.8) | 163 (25.2) | 0 (0) | 225 (64.3) | 20.0 | 80.0 |
| **Total** | | **1346** | **573 (42.6)** | **189 (14)** | **584 (43.4)** | **1022 (75.9)** | **320 (23.8)** | **4 (0.3)** | **525 (69.3)** | **26.5** | **73.5** |
| SKEL | bst | 19 | 19 (100) | 0 (0) | 0 (0) | 14 (73.7) | 5 (26.3) | 0 (0) | 14 (73.7) | 20.0 | 80.0 |
| | colorsys | 8 | 8 (100) | 0 (0) | 0 (0) | 7 (87.5) | 1 (12.5) | 0 (0) | 7 (87.5) | 0.0 | 100 |
| | heapq | 22 | 19 (86.4) | 3 (13.6) | 0 (0) | 12 (54.5) | 10 (45.5) | 0 (0) | 13 (59.1) | 50.0 | 50.0 |
| | html | 44 | 40 (90.9) | 2 (4.5) | 2 (4.5) | 35 (79.5) | 9 (20.5) | 0 (0) | 33 (78.6) | 66.7 | 33.3 |
| | mathgen | 81 | 77 (95.1) | 4 (4.9) | 0 (0) | 65 (80.2) | 16 (19.8) | 0 (0) | 67 (82.7) | 50.0 | 50.0 |
| | rbt | 27 | 26 (96.3) | 0 (0) | 1 (3.7) | 23 (85.2) | 4 (14.8) | 0 (0) | 22 (84.6) | 75.0 | 25.0 |
| | strsim | 64 | 50 (78.1) | 0 (0) | 14 (21.9) | 56 (87.5) | 8 (12.5) | 0 (0) | 44 (88) | 40.0 | 60.0 |
| | toml | 72 | 37 (51.4) | 10 (13.9) | 25 (34.7) | 49 (68.1) | 22 (30.6) | 1 (1.4) | 33 (71.7) | 40.0 | 60.0 |
| **Total** | | **337** | **276 (81.9)** | **19 (5.6)** | **42 (12.5)** | **261 (77.4)** | **75 (22.3)** | **1 (0.3)** | **233 (79.3)** | **46.5** | **53.5** |
| RUSTREPOTRANS | charset | 33 | 20 (60.6) | 13 (39.4) | 0 (0) | 14 (42.4) | 19 (57.6) | 0 (0) | 25 (75.8) | 75.0 | 25.0 |
| | deltachat | 125 | 54 (43.2) | 69 (55.2) | 2 (1.6) | 39 (31.2) | 84 (67.2) | 2 (1.6) | 92 (76) | 90.0 | 10.0 |
| | iceberg-java | 25 | 9 (36) | 16 (64) | 0 (0) | 3 (12) | 16 (64) | 6 (24) | 15 (78.9) | 100 | 0.0 |
| | iceberg-py | 44 | 15 (34.1) | 28 (63.6) | 1 (2.3) | 11 (25) | 29 (65.9) | 4 (9.1) | 35 (89.7) | 100 | 0.0 |
| | crypto-c | 20 | 16 (80) | 4 (20) | 0 (0) | 7 (35) | 13 (65) | 0 (0) | 11 (55) | 66.7 | 33.3 |
| | crypto-java | 97 | 39 (40.2) | 58 (59.8) | 0 (0) | 30 (30.9) | 67 (69.1) | 0 (0) | 86 (88.7) | 100 | 0.0 |
| **Total** | | **344** | **153 (44.5)** | **188 (54.7)** | **3 (0.9)** | **104 (30.2)** | **228 (66.3)** | **12 (3.5)** | **264 (80.2)** | **87.5** | **12.5** |
| **Total** | | **2219** | **1140 (51.4)** | **449 (20.2)** | **630 (28.4)** | **1519 (68.5)** | **682 (30.7)** | **18 (0.8)** | **1143 (72.8)** | **39.3** | **60.7** |

rect patches to repair translation bugs. SKEL (Wang et al., 2025a) reprompts the same LLM for repairing bugs, but then requires a user to manually provide a fix. ALPHA-TRANS (Ibrahimzada et al., 2025) and OXIDIZER (Zhang et al., 2025) generates patches in the loop, however, no effectiveness was reported in their papers, and we could not repair any translation bugs using their tools. Moreover, patches by ALPHATRANS and OXIDIZER could not be validated mostly because of limitations in their validation system. For example, the following code snippets show instance 11 from the project commons-validator in ALPHATRANS. The GraalVM-based validation system in ALPHATRANS does not validate this translation as functionally equivalent, although our manual investigation indicates that the Python translation is correct. Therefore, the limitation in ALPHA-TRANS is mostly due to its validation system being unable to validate LLM patches.

```
------------------- JAVA SOURCE CODE -------------------
public boolean isOn(long flag) {
    return (this.flags & flag) == flag;
}
```

```
------------------- PYTHON TRANSLATION -------------------
def isOn(self, flag: int) -> bool:

    return (self.__flags & flag) == flag
```

Column MATCHFIXAGENT *Repaired* shows the number of translation bugs successfully repaired by our approach. In total, MATCHFIXAGENT repaired $\frac{134}{265}$ (50.6%) of translation bugs, 32.1% more than existing reprompting-based techniques. Given the limitations in validation system of ALPHATRANS discussed earlier, we manually

investigate and validate patches from this tool [3]. The following code snippets demonstrate instance 276 from the project commons-validator in ALPHATRANS in which its reprompting-based repairing could not generate a correct patch. By contrast, MATCHFIXAGENT successfully repairs this translation bug with the help of its Library Analyzer (§2.2.4). The report generated by this semantic analyzer indicate "... the standard Python datetime.date class does not have a SHORT attribute ..." which is correct. The Test Generator & Repair Agent (§2.3) then leverages this analysis and successfully generate a patch by replacing SHORT with constant 3.

```
------------------- JAVA SOURCE CODE -------------------
public static DateValidator DateValidator1() {
    return new DateValidator(true, DateFormat.SHORT);
}
```

```
------------------- PYTHON TRANSLATION -------------------
def DateValidator1() -> DateValidator:
-    return DateValidator(True, datetime.date.SHORT)
+    return DateValidator(True, 3)
```

Column *Disagreement Repaired* shows the number of disagreements from RQ1 (§3.1) which MATCHFIXAGENT determined as *not equivalent* and successfully generated a correct patch. Of the 49 disagreements resolved in favor of MATCHFIXAGENT, we further asked our manual investigators if the generated patch was correct or not. On average, $\frac{47}{49}$ (95.9%) of patches were validated as correct. Notice

[3]Please refer to our artifacts website (Intelligent-CAT-Lab, 2026) for detailed investigation results.

*Table 2.* Effectiveness of MATCHFIXAGENT in translation repair compared against existing techniques. **NEQ:** Not Equivalent, **NR:** Not Reported/Repaired.

| Tool | Project | Total # Trans. Pair | Tool NEQ ∩ MATCHFIXAGENT NEQ | Tool Repaired | MATCHFIXAGENT Repaired | Disagreement Repaired | Coverage (Improvement) % |
|---|---|---|---|---|---|---|---|
| OXIDIZER | checkdigit | 29 | 5 (17.2) | NR | 5 (100) | 2 (100) | 86.2 (0) |
| | go-edlib | 24 | 2 (8.3) | NR | 1 (50) | 4 (100) | 100 (0) |
| | histogram | 19 | 2 (10.5) | NR | 1 (50) | 2 (66.7) | 68.4 (0) |
| | nameparts | 15 | 3 (20) | NR | 1 (33.3) | 0 (0) | 100 (0) |
| | stats | 53 | 6 (11.3) | NR | 6 (100) | 5 (100) | 100 (↑**1.9**) |
| | textrank | 52 | 3 (5.8) | NR | 3 (100) | 2 (100) | 100 (0) |
| **Total** | | **192** | **21 (10.9)** | **0 (0)** | **17 (81)** | **15 (93.8)** | **92.4 (↑0.3)** |
| ALPHATRANS | cli | 273 | 9 (3.3) | NR | 7 (77.8) | 3 (100) | 100 (↑**6.2**) |
| | csv | 235 | 20 (8.5) | NR | 16 (80) | 2 (100) | 100 (↑**11.9**) |
| | fileupload | 192 | 0 (0) | - | - | 2 (100) | 98.9 (↑**78.1**) |
| | validator | 646 | 34 (5.3) | NR | 23 (67.6) | 3 (100) | 99.3 (↑**36.8**) |
| **Total** | | **1346** | **63 (4.7)** | **0 (0)** | **46 (73)** | **10 (100)** | **99.6 (↑33.3)** |
| SKEL | bst | 19 | 0 (0) | - | - | 4 (100) | 100 (0) |
| | colorsys | 8 | 0 (0) | - | - | 1 (100) | 100 (0) |
| | heapq | 22 | 2 (9.1) | NR | 1 (50) | 3 (100) | 100 (0) |
| | html | 44 | 1 (2.3) | NR | 0 (0) | 2 (100) | 100 (↑**4.5**) |
| | mathgen | 81 | 3 (3.7) | NR | 1 (33.3) | 2 (66.7) | 100 (0) |
| | rbt | 27 | 0 (0) | - | - | 1 (100) | 100 (↑**3.7**) |
| | strsim | 64 | 0 (0) | - | - | 3 (100) | 100 (↑**21.9**) |
| | toml | 72 | 5 (6.9) | NR | 3 (60) | 3 (100) | 100 (↑**34.7**) |
| **Total** | | **337** | **11 (3.3)** | **0 (0)** | **5 (45.5)** | **19 (95)** | **100 (↑8.1)** |
| RUSTREPOTRANS | charset | 33 | 12 (36.4) | 5 (41.7) | 7 (58.3) | 1 (100) | 100 (0) |
| | deltachat | 125 | 60 (48) | 10 (16.7) | 11 (18.3) | 1 (100) | 100 (↑**1.6**) |
| | iceberg-java | 25 | 12 (48) | 1 (8.3) | 1 (8.3) | 0 (0) | 100 (0) |
| | iceberg-py | 44 | 25 (56.8) | 4 (16) | 6 (24) | 0 (0) | 100 (↑**2.3**) |
| | crypto-c | 20 | 4 (20) | 1 (25) | 2 (50) | 1 (100) | 100 (0) |
| | crypto-java | 97 | 57 (58.8) | 28 (49.1) | 39 (68.4) | 0 (0) | 100 (0) |
| **Total** | | **344** | **170 (49.4)** | **49 (28.8)** | **66 (38.8)** | **3 (100)** | **100 (↑0.6)** |
| **Total** | | **2219** | **265 (11.9)** | **49 (18.5)** | **134 (50.6)** | **47 (95.9)** | **98.1 (↑ 8.5)** |

*Table 3.* Development cost of MATCHFIXAGENT compared against existing tools.

| Tool | Complexity (LoC) | Development Time (Months) |
|---|---|---|
| ALPHATRANS | 10859 | 8 |
| OXIDIZER | 19052 | 10 |
| SKEL | 3843 | 6 |
| MATCHFIXAGENT (ours) | 1650 | 0.7 |

that this column cannot be directly compared with existing techniques, because they validated disagreements as functionally correct and did not generate any patches. Moreover, Column *Coverage* indicates the overall coverage for subject projects and the total improvement as a result of tests generated by MATCHFIXAGENT. The generated tests help improve code coverage by 8.5% (from 89.6% to 98.1%). Project `commons-fileupload` from ALPHATRANS sees the most improvement (78.1%), with most of its translated fragments now validated with MATCHFIXAGENT tests.

### 3.3. Development Cost and Adaptability

#### 3.3.1. DEVELOPMENT COST

Table 3 shows the development cost of MATCHFIXAGENT against existing techniques. We define cost as the tool's total lines of code (LoC). As shown in the table, developing MATCHFIXAGENT is cheap and the initial version only consists of 1,650 LoC, supporting 6 different PLs. Its dependence only on the Tree-Sitter (Tree-Sitter, 2026) parser makes it easy to support more languages using only 280 LoC. By contrast, other tools, such as, SKEL (Wang et al., 2025a), ALPHATRANS (Ibrahimzada et al., 2025), and OXIDIZER (Zhang et al., 2025) only support *one* PL pair and require significant engineering effort to adapt to more languages. More precisely, MATCHFIXAGENT is ×2.3, ×6.6, and ×11.6 cheaper than SKEL, ALPHATRANS, and OXIDIZER, respectively. The static nature of MATCHFIXAGENT makes it cost-effective and scalable, achieving better performance and revealing major limitations in existing tools.

#### 3.3.2. ADAPTABILITY

The architecture of MATCHFIXAGENT is largely independent of any specific LLM or agent framework, making it easy to extend and integrate with more LLMs. In this research question, we investigate the extent to which replacing Anthropic's Claude 3.7 Sonnet with OpenAI `o4-mini-2025-04-16` (OpenAI, 2025), and Claude Code with Codex (OpenAI, 2026a) agent, impacts the performance of MATCHFIXAGENT. Due to the limited cost budget, we randomly sampled 96 instances from all subject projects. While sampling, we controlled for equal contribution of equivalent and non-equivalent translations, resulting in 58 and 38 samples for each, respectively.

Figure 2 illustrates the results of this study. Our analysis indicates that MATCHFIXAGENT with Claude Code and Codex agrees 73% of the time against existing validation systems. Moreover, we also analyzed both agents' behavior in terms of problem understanding and finding a solution. Our investigation of agent trajectories (footprint of agent

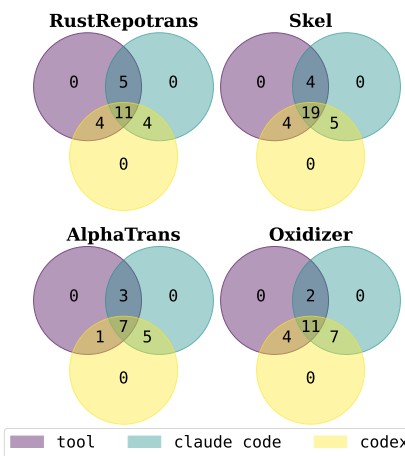

*Figure 2.* Agreement and dispute cases between tool validation system and MATCHFIXAGENT with Claude and Codex agents.

actions) shows OpenAI's Codex makes fewer actions to explore the codebase, and attempts to provide a decision faster. By contrast, Anthropic's Claude Code agent first plans and reasons thoroughly about its tasks, specifically called `TODO List` by the agent, and then takes specific actions to perform each of its planned tasks.

### 3.4. Ablation Study

#### 3.4.1. IMPACT OF SEMANTIC ANALYZER AND TEST GENERATOR AGENT

To investigate the importance of semantic analyzer and test generator agent in MATCHFIXAGENT, we evaluated a standalone baseline agent that uses the same LLM and agent framework, e.g., `Claude Sonnet 3.7` (Anthropic, 2026b) and Claude Code (Anthropic, 2026c) [4]. In order to perform a controlled study, we created a sample from the original dataset where existing tools and MATCHFIXAGENT verdicts agree with each other, meaning we collected all non-dispute instances. In total, 1,091 instances, 862 equivalent, and 229 non-equivalent translations were selected. Figure 3 shows the result of this ablation study. Accuracy indicates the ratio of baseline verdicts that agree with the existing tool and MATCHFIXAGENT. On average, the validation accuracy drops by 42.3%, illustrating the importance of MATCHFIXAGENT's semantic analyzer and test generator components. Across all projects, the baseline agent only reproduced MATCHFIXAGENT and tool results in the `histogram` project from OXIDIZER (Zhang et al., 2025), achieving 100% validation correctness. For the remaining projects, the agent's accuracy dropped as low as 25% in `iceberg-java`, which is the project with the largest number of lines of code.

Moreover, we also evaluated our baseline agent on dispute

---

[4]Please refer to our artifacts (Intelligent-CAT-Lab, 2026) for prompt templates used in this study.

instances, a total of 416 cases between MATCHFIXAGENT and existing tools. The results indicate 48.3% agreement with MATCHFIXAGENT, and 41.1% with competing tools, however, these numbers do not convey any important meaning without ground-truths.

To address this issue, we considered the 145 cases where manual investigation established ground truth (§D.1), and we can directly measure baseline agent accuracy. On the 145 manually resolved cases, the baseline agent achieves only 47.7% accuracy on the 88 cases where MATCHFIXAGENT is correct–the cases that represent MATCHFIXAGENT's unique advantage over existing tools. By contrast, the baseline achieves 56.1% on the 57 cases where the existing tool is correct, confirming that the baseline agent systematically fails on precisely the hard, ambiguous translations that MATCHFIXAGENT 's semantic analyzer and test generation are designed to handle.

The 47.7% accuracy on difficult, disputed cases (where our original ablation showed only a 42.3% drop on easier, non-disputed cases) confirms that MATCHFIXAGENT's multi-agent architecture is essential, not just helpful. The baseline agent's near-random performance on these difficult cases demonstrates that the semantic analyzer and test generation components are the core drivers of MATCHFIXAGENT's accuracy advantage in resolving both straightforward and challenging translations.

#### 3.4.2. IMPACT OF SEMANTIC ANALYZER

We perform another study by removing only the semantic analysis results when prompting the test generator and verdict agents. We use the same set with 1,091 instances from the previous ablation to perform this study. Figure 4 illustrates the result of our second ablation. The results indicate that the performance of MATCHFIXAGENT significantly drops by 39.7% without the six semantic analyses. Furthermore, we observe that the test generator agent without semantic analyzer is more costly and on average spends 3.9% ($66.3K$ instead of $63.7K$), 6.2% ($136K$ instead of $128K$), 4.2% ($9.4M$ instead of $9.0M$), and 7.5% ($61.28\,\mathrm{h}$ instead of $56.71\,\mathrm{h}$) more turns/interactions, input tokens, output tokens, and time, respectively.

## 4. Related Work

**Translation Validation and Repair.** Existing automated translation validation techniques either rely on test execution (Pan et al., 2024; Zhang et al., 2025; Ibrahimzada et al., 2025; Shetty et al., 2024; Ou et al., 2025; Ziftci et al., 2025; Xue et al., 2025; Yang et al., 2024b; Ke et al., 2025; Dehghan et al., 2025; Ibrahimzada et al., 2026; Roziere et al., 2020; Khatry et al., 2025), formal methods (Yang et al., 2025; Nitin et al., 2024; Garzella et al., 2020), or fuzzing (Eniser

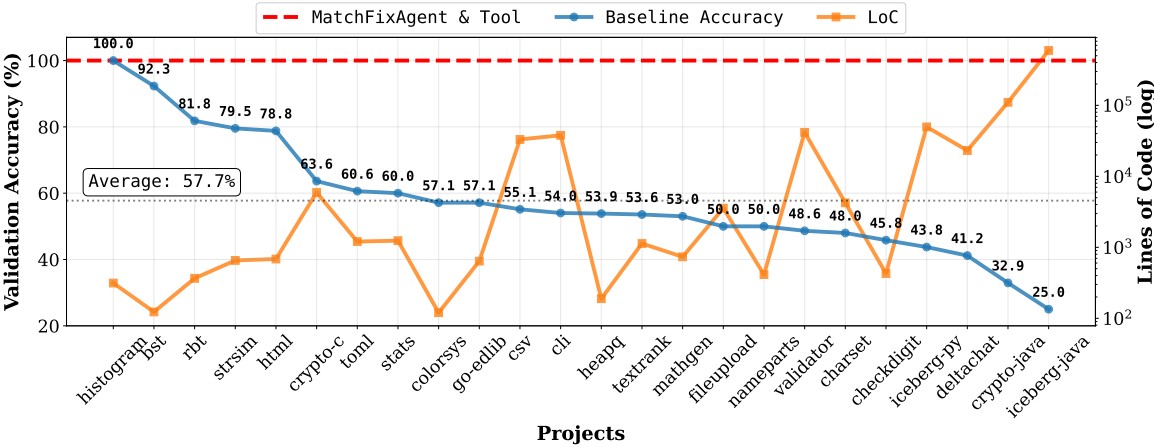

*Figure 3.* Impact of semantic analyzer and test generator agent in MATCHFIXAGENT. A standalone baseline agent struggles validating translations as the projects grow in size.

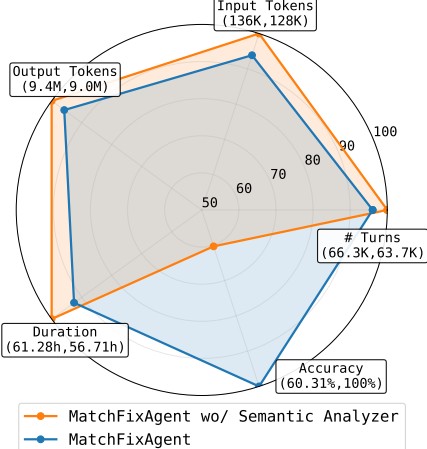

*Figure 4.* Removing the semantic analyzer decreases the effectiveness of MATCHFIXAGENT, while increasing token consumption, number of turns, and processing time.

et al., 2024) for translation validation. Abid et al. (Abid et al., 2024) and ALPHATRANS (Ibrahimzada et al., 2025) leverage GraalVM (Oracle, 2026) and language interoperability to execute code in both source and target PL for translation validation. Other tools like OXIDIZER (Zhang et al., 2025) and SYZYGY (Shetty et al., 2024) instrument programs to extract input-output pairs and use in target PL for validation. SKEL (Wang et al., 2025a) validates translations through test execution by converting Python tests to JavaScript simply through string replacement. This suffices since source Python tests are simple function calls and value comparisons. For automated translation repair, most tools (Zhang et al., 2025; Wang et al., 2025a; Ibrahimzada et al., 2025; Shetty et al., 2024; Ou et al., 2025; Yang et al., 2024b; Pan et al., 2024) rely on simple reprompting of LLMs with execution feedback, which has proven ineffective. Specifically, ALPHATRANS (Ibrahimzada et al., 2025) performs independent reprompting of suspicious fragments based on execution trace, while SKEL (Wang et al., 2025a)

requires a user for manually repairing translation bugs.

**LLM Agents.** With the increasing prominence of agent-based frameworks, recent research and industrial efforts have turned towards leveraging these frameworks to address various software engineering tasks. SWE-agent (Yang et al., 2024a) introduces a specialized agent-computer interface (ACI), facilitating agent interaction with code repositories via file reading, editing, and execution of bash commands. AUTOCODEROVER (Zhang et al., 2024), which provides LLM agents with specialized code-search APIs, enables iterative retrieval and localization of code segments associated with bugs. SPECROVER (Ruan et al., 2025) enhances AUTOCODEROVER by emphasizing specification inference, generating function summaries, and providing targeted feedback at crucial agent execution stages. Besides these state-of-the-art frameworks, numerous additional agent-based approaches exist both in open-source (Wei et al., 2025b; Ouyang et al., 2025; Bouzenia et al., 2025; Team, 2026d; AI, 2026) and commercial products (Wang et al., 2025b; Amazon, 2026; Cognition, 2026).

## 5. Conclusion

We presented MATCHFIXAGENT, a *language-agnostic* technique that combines the power of program analysis and LLM agents for autonomous repository-level code translation validation and repair. It performs various semantic analyses to systematically generate targeted tests, enabling demonstration of functional equivalence or detection of semantic bugs. Through rigorous evaluation on multiple benchmarks and different PL-pairs, we show the effectiveness of MATCHFIXAGENT. Our manual investigation of generated reports reveals significant limitation of existing techniques. To our knowledge, MATCHFIXAGENT is the first approach that can effectively validate and repair translations in repository-level across multiple PLs.

## Acknowledgements

We thank the cohort of Summer 2025 interns and mentors at AWS for their valuable feedback. We also thank the anonymous reviewers for their comments, which helped make this work stronger, and Professor Elsa Gunter for partially funding one of the authors during this research.

## Impact Statement

This work is motivated to boost large language model agents in repository-level code translation validation and repair across multiple programming languages. Software developers spend a significant amount of time migrating code and testing them. We believe our tool will significantly improve the developers' experience in their day-to-day life. We envision two ways developers can use MATCHFIXAGENT in their workflow. First, stakeholders can first migrate their codebase, and run MATCHFIXAGENT to clarify further requirements and detect missing corner-case functionality. Second, MATCHFIXAGENT can be directly integrated with existing and new automated code translation tools to provide a second level of confidence in translation validation.

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

## A. Software and Data

The implementation of MATCHFIXAGENT, the results of our human study, the agent logs and trajectories, and the docker images required for reproducing the results presented in this paper are publicly available (Intelligent-CAT-Lab, 2026).

## B. Limitations of Prior Work

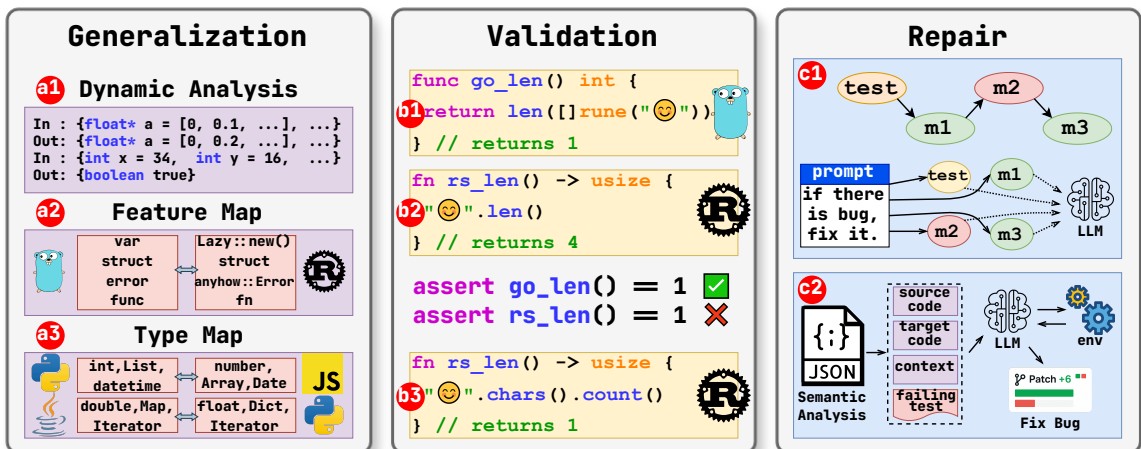

*Figure 5.* Illustration of key limitations of existing techniques in validation and repair of repository-level code translation and MATCHFIX-AGENT addressing them.

To demonstrate the limitations of existing techniques (Ibrahimzada et al., 2025; Zhang et al., 2025; Wang et al., 2025a; Shetty et al., 2024) for validation and repair, we use the examples in Figure 5.

**Limitation 1: Generalization.** Existing automated translation tools require substantial engineering effort to build validation systems for individual language pairs. For instance, OXIDIZER (Zhang et al., 2025) and SYZYGY (Shetty et al., 2024) require dynamic analysis and I/O extraction from source code (a1). OXIDIZER further requires a predefined feature map (a2), which together with their dynamic analyzer is 19,000 lines of code. Other tools like ALPHATRANS (Ibrahimzada et al., 2025) and SKEL (Wang et al., 2025a) validate programs using high quality type map (a3), which requires manual maintenance over time as PLs evolve. In comparison, MATCHFIXAGENT uses only language-agnostic LLM prompts, an off-the-shelf LLM coding agent tool, and lightweight static analysis. While the static analysis must be implemented for each PL, each PL requires approximately 280 additional lines of code to support, hence it is extremely easy to generalize to many PLs.

**Limitation 2: Validation.** Most prior work depends on existing source project tests for translation validation, but these tests may have insufficient coverage. The test suites in ALPHATRANS (Ibrahimzada et al., 2025) have an average of $56.57\%$ method coverage. Even when developer-written tests provide adequate coverage, they may miss the edge cases that expose subtle semantic differences between source and target languages. Consider the real-world case from OXIDIZER (Zhang et al., 2025), where a Go program (b1) that counts characters in a string is translated to Rust (b2) using the .len() method, which counts bytes rather than characters. OXIDIZER validates this pair as *functionally equivalent* because it only exercises this program with ASCII inputs. MATCHFIXAGENT, in contrast, marks the translation as not equivalent, and synthesizes a test with Unicode inputs (e.g., U+1F60A, a four-byte emoji representing a single character) to confirm the inequivalence. Upon detecting the translation bug, it automatically generates a patch that uses .chars().count() to ensure proper handling of both ASCII and Unicode characters (b3).

**Limitation 3: Repair.** Current translation repair techniques solely rely on simple feedback-driven approaches, which have proven inadequate in practical settings. SKEL (Wang et al., 2025a), OXIDIZER (Zhang et al., 2025), and SYZYGY (Shetty et al., 2024) utilize multi-turn iterative prompting techniques. ALPHATRANS (Ibrahimzada et al., 2025) adopts an execution trace-based reprompting strategy. For instance, consider scenario (c1), where running the test method sequentially invokes methods m1, m2, and m3. If a bug is present in fragment m2, ALPHATRANS reprompts all executed fragments individually without considering inter-fragment dependencies. This approach, while potentially resolving the bug in m2, risks introducing new functional bugs in previously correct fragments, such as m1. To overcome this limitation, MATCHFIXAGENT uses code analysis and an LLM agent (c2). The analyses expose dependencies such as the one between m1 and m2 and the agent interacts with the execution environment to execute, validate, and iteratively refine its generated patches.

*Table 4.* Details of benchmarks[5] from existing techniques used in MATCHFIXAGENT. **LoC:** Lines of code in the source project.

| Tool | Project | Source Language | Target Language | Total # Trans. Pairs | LoC | Test Coverage (%) | Stars | Forks |
|------|---------|-----------------|-----------------|----------------------|-----|-------------------|-------|-------|
| OXIDIZER (Zhang et al., 2025) | checkdigit (Tonomori, 2026) | Go | Rust | 29 | 428 | 86.2 | 111 | 8 |
| | go-edlib (Bollon, 2026) | | | 24 | 639 | 100 | 517 | 27 |
| | histogram (Cortex, 2026) | | | 19 | 314 | 68.4 | 176 | 31 |
| | nameparts (Polera, 2026) | | | 15 | 413 | 100 | 43 | 5 |
| | stats (Flynn, 2026) | | | 53 | 1241 | 98.1 | 2989 | 170 |
| | textrank (Belicza, 2026) | | | 52 | 1132 | 100 | 217 | 22 |
| ALPHATRANS (Ibrahimzada et al., 2025) | cli (Foundation, 2026a) | Java | Python | 273 | 37841 | 93.8 | 372 | 201 |
| | csv (Foundation, 2026b) | | | 235 | 33072 | 88.1 | 392 | 278 |
| | fileupload (Foundation, 2026c) | | | 192 | 3567 | 20.8 | 246 | 185 |
| | validator (Foundation, 2026d) | | | 646 | 41605 | 62.5 | 216 | 164 |
| SKEL (Wang et al., 2025a) | bst (Algorithms, 2026a) | Python | JavaScript | 19 | 123 | 100 | 203000 | 47000 |
| | colorsys (Python, 2026a) | | | 8 | 120 | 100 | 67900 | 32300 |
| | heapq (Python, 2026c) | | | 22 | 189 | 100 | 67900 | 32300 |
| | html (Python, 2026d) | | | 44 | 684 | 95.5 | 67900 | 32300 |
| | mathgen (Weiler, 2026) | | | 81 | 735 | 100 | 711 | 183 |
| | rbt (Algorithms, 2026b) | | | 27 | 366 | 96.3 | 203000 | 47000 |
| | strsim (Luo, 2026) | | | 64 | 654 | 78.1 | 1014 | 125 |
| | toml (Pearson, 2026) | | | 72 | 1206 | 65.3 | 1126 | 192 |
| RUSTREPOTRANS (Ou et al., 2025) | charset (Jawah, 2026) | Python | Rust | 33 | 4231 | 100 | 672 | 56 |
| | deltachat (Chat, 2026) | C | | 125 | 23116 | 98.4 | 306 | 28 |
| | iceberg-java (Foundation, 2026g) | Java | | 25 | 592793 | 100 | 7700 | 2700 |
| | iceberg-py (Foundation, 2026h) | Python | | 44 | 49746 | 97.7 | 805 | 327 |
| | crypto-c (Foundation, 2026e) | C | | 20 | 5922 | 100 | 36 | 15 |
| | crypto-java (Foundation, 2026f) | Java | | 97 | 110261 | 100 | 2 | 7 |
| **Total** | | | | **2219** | **910398** | **89.6** | **627351** | **195624** |

## C. Experimental Setup

### C.1. Benchmark

We evaluate MATCHFIXAGENT on benchmarks used in prior work on automated repository-level translation. Each benchmark problem is a translation pair: the source function and the corresponding translation. The task for each benchmark problem is to give a verdict on the functional equivalence of pairs in two different PLs, and repair translation in case of equivalence. Our subjects are open-source repository-level translations with equivalence verdicts available[6] from the peer-reviewed literature[7]. We make selections across a diverse set of PL pairs.

Table 4 summarizes our subject translation pairs. We collect subjects from three recent repository-level code translation techniques (Ibrahimzada et al., 2025; Zhang et al., 2025; Wang et al., 2025a). We did not include SYZYGY (Shetty et al., 2024) because its artifact does not provide a validation system for individual functions (it only provides end-to-end tests). These works generated translations of real-world open source GitHub projects, and performed equivalence validation at the individual function level. Since ALPHATRANS (Ibrahimzada et al., 2025) is evaluated on a large number of functions, we randomly sample 1346 of its total 4643 translation pairs[8].

To demonstrate the adaptability of MATCHFIXAGENT to more PL pairs, we also collect subjects from RUSTREPOTRANS (Ou et al., 2025), a benchmark consisting of human-written translations into Rust and unit tests. We exclude translation pairs collected from the libp2p (Libp2p, 2026) projects in RUSTREPOTRANS due to the presence of non-deterministic flaky tests that may result in false negatives, i.e., functional inequivalence while translation is correct, unfairly biasing comparison in favor of MATCHFIXAGENT. In total, we collect 2,219 translation pairs with over $900K$ lines of code from 24 projects and in 6 different PL pairs.

### C.2. LLMs

Major software engineering leaderboards (Team, 2026f;a) have shown that Claude Sonnet (Anthropic, 2026b) outperforms other proprietary LLMs, such as OpenAI GPT-4o (OpenAI, 2026b) and Google Gemini Pro (Deepmind, 2026). Therefore,

---

[5]Some projects in SKEL (e.g., `colorsys`) are part of a bigger project (Python, 2026b). The reported Star and Fork numbers belong to that bigger project.

[6]We exclude rule-based transpilers as they do not include a validation mechanism, i.e., their translation is (theoretically) correct by construction.

[7]This criterion excludes techniques such as RustMap (Cai et al., 2025) and C2SaferRust (Nitin et al., 2026).

[8]ALPHATRANS translated both application and test code. In this work, we only included the application code translation pairs. While reviewing their artifacts, we also noted 11 translation pairs to be dead code and excluded them.

we use Anthropic's `Claude 3.7 Sonnet` (Anthropic, 2026b) and Claude Code (1.0.51) (Anthropic, 2026c) as the main LLM and agent in all our experiments. To show the adaptability of MATCHFIXAGENT to different LLMs and agentic frameworks, we repeat a subset of experiments using OpenAI `o4-mini-2025-04-16` (OpenAI, 2025) and Codex (OpenAI, 2026a) (§3.3). To support future research and external validation, MATCHFIXAGENT logs the inputs, intermediate agent interactions, tool execution results, and outputs of the LLM, and supports visualizing and inspecting these logs. MATCHFIX-AGENT terminates within the budget of 1,000 seconds. We empirically set this timeout after analyzing the execution time of 300 samples.

### C.3. Competing Validation & Repair Tools

We compare MATCHFIXAGENT's validation technique with the automated validation techniques proposed by SKEL (Wang et al., 2025a), OXIDIZER (Zhang et al., 2025), and ALPHATRANS (Ibrahimzada et al., 2025). Except RUSTREPOTRANS, other approaches do not explicitly report the repair results, as the repair process is interleaved with translation in the loop. For those techniques (Ibrahimzada et al., 2025; Zhang et al., 2025; Wang et al., 2025a), we compare MATCHFIXAGENT repair results with their final translation success.

### C.4. MATCHFIXAGENT Implementation

We implement our structure-based semantic analysis on top of Tree-Sitter (Tree-Sitter, 2026), as it supports 165+ languages, including six PLs we target in this study. For running tests and validating patches, MATCHFIXAGENT uses Rust 1.87.0 (RustLang, 2026), Python 3.12.9 (Python, 2026e), Java 21.0.7 (Java, 2026), Node 22.16.0 (Team, 2026e), GCC 7.3.1 (Team, 2026b), and Go 1.24.4 (GoLang, 2026).

## D. Analysis of (Dis)Agreements and Examples of Incorrect Verdicts for Claude Experiments

### D.1. Analysis of the Disagreements

Both MATCHFIXAGENT and existing validation approaches are prone to false positives (i.e. the verdict is equivalent but the translation is not) and false negatives. To determine false positives and false negatives, we perform a manual investigation.

We first categorize disagreements into two cases: $D_1$, where MATCHFIXAGENT produced a *not equivalent* verdict and the other disagreed; and $D_2$, where MATCHFIXAGENT produced an *equivalent* verdict and the other disagreed. Due to large number of studied translation pairs, we randomly sample five instances of both $D_1$ and $D_2$ from each of the 24 projects. If a project had fewer than five disagreements, we considered all instances without sampling. Two authors[9] independently reviewed disagreements to verify whether *the not equivalent verdict* was correct (by MATCHFIXAGENT in $D_1$ and by others in $D_2$) and achieved an inter-rater agreement of 83.7%. If the not equivalent verdict is correct, the reviewer rules it in favor of the tool that said not equivalent, otherwise they rule it favor of the tool that said equivalent. During the investigation, 3 disagreements with OXIDIZER were due to the tool's function mocking not being enabled, and were unable to enable it. In addition, 11 disagreements with RUSTREPOTRANS were due to the correct translation not being "1:1", or in other words, the correct translation is not functionally equivalent to the source function. These disagreements would have been unfairly ruled in favor of MATCHFIXAGENT, and so were filtered out, leaving us with 145 disagreement cases ($D_1 = 92$, $D_2 = 53$). When the reviewer's resolution conflicted (18.6% of cases), they met with each other and agreed on the final resolution[10].

The **Disagreement** columns in Table 1 summarize the result of our human investigation. The **Tool** column shows the percentage of disagreements ruled in favor of the existing validation tool, and the **Ours** columns shows the percentage ruled in favor of MATCHFIXAGENT. The disagreement resolutions show that MATCHFIXAGENT 's verdicts are often more accurate than existing automated validation tools. MATCHFIXAGENT's verdicts are significantly more accurate than OXIDIZER and ALPHATRANS— disagreements are ruled in favor of MATCHFIXAGENT in 84.1% and 73.5% of cases, respectively. Compared to SKEL, MATCHFIXAGENT shows similar accuracy (53.5%). On RUSTREPOTRANS, MATCHFIXAGENT's accuracy fares worse (12.5%), which is due to the relative complexity of its translation pairs.

Existing validation approaches produced 88 incorrect verdicts, 80 of which fell into three categories: (1) **Inadequate Unit Tests (42 of cases)**, (2) **Excessively Strict Equivalence Definition (11 of cases)**, and (3) **Language Interoperability Bug (27 of cases)**. A language interoperability bug means that the tool's process for converting concrete inputs in the source

---

[9]The selected authors have total experience of 17 years in academia and 4.5 years in industry.

[10]The results of our human investigation with comments by each reviewer are publicly available (Intelligent-CAT-Lab, 2026).

language to the target language did not preserve equivalence. For example, OXIDIZER's conversion of `Go runes` (which represent a Unicode character) to `Rust chars` resulted in different Unicode characters.

On the benchmarks associated with automated validation tools (SKEL, OXIDIZER, ALPHATRANS), MATCHFIXAGENT produced 36 incorrect verdicts, 34 of which fell into three categories: (1) **Hallucination (23 cases)**, (2) **Inadequate Unit Tests (4 cases)** (meaning MATCHFIXAGENT missed an input that would demonstrate inequivalence), (3) **Infeasible Input (7 cases)**. An infeasible input means that the LLM discovered an input where the source function and translation produce different outputs, but the input can never occur when the project is used as intended. Infeasible inputs often involve directly initializing private class/struct members, or calling private helper methods in unintended ways.

On RUSTREPOTRANS's benchmarks, MATCHFIXAGENT produced 21 incorrect verdicts. Two of the main causes are similar to the other benchmarks: (1) **Hallucination (7 cases)** and (2) **Inadequate Unit Tests (6 cases)**. The increased rates of these two causes are due to the relative complexity and size of RUSTREPOTRANS's projects. The other major cause is **Excessively Strict Equivalence Definition (6 cases)**. As previously mentioned, RUSTREPOTRANS's translations include refactors to make the translation more idiomatic, which creates ambiguity around the proper definition equivalence.

### D.2. Representative Examples of Incorrect Verdicts

The following code snippet shows an example of **Inadequate Unit Tests** on a translation pair from OXIDIZER. The functions both calculate the edit distance between two strings. The functions are not equivalent because Go's `len()` function counts Unicode characters, whereas Rust's `.len()` function counts bytes. OXIDIZER incorrectly validates the Rust translation as equivalent because the source project's unit tests do not cover non-ASCII inputs. However, MATCHFIXAGENT successfully generates a test with non-ASCII inputs demonstrating they are not equivalent.

```
1 ------------------ GO SOURCE CODE ------------------
2 func LCSEditDistance(str1 string, str2 string)
↪    int {
3
4     // ... if conditions ...
5     lcs := LCS(s1, s2)
6     return (len([]rune(s1)) - lcs) +
↪        (len([]rune(s2)) - lcs)
7 }
```

```
1 ------------------ RUST TRANSLATION ------------------
2 pub fn lcs_edit_distance(str1: &str, str2: &str) ->
↪    Result<i32> {
3     // ... if conditions ...
4     let lcs_len = lcs(str1, str2)?;
5     let edit_distance = (str1.len() as i32 - lcs_len)
↪        + (str2.len() as i32 - lcs_len);
6     Ok(edit_distance)
7 }
```

The next code snippet demonstrates an example of an **Infeasible Input** taken from the `textrank` project. The below Go function inserts a value into the map `ranks.SentenceMap` (respectively, `ranks.sentence_map` for the Rust translation). MATCHFIXAGENT discovers that these functions return different values when the map is initially `{ 1 : "SomeString" }` (the Go returns 0 while the Rust returns 1). However, when the `textrank` is used properly via its public interface, this initial state for the map cannot occur. The map will always contain keys from 1 to $n$, where $n$ is the number of map entries. Under this precondition, the functions are equivalent.

```
1 ------------------ GO SOURCE CODE ------------------
2 func addSentence(ranks *Rank, sentence
↪    ParsedSentence) int {
3     ranks.SentenceMap[len(ranks.SentenceMap)] =
↪        sentence.GetOriginal()
4     return len(ranks.SentenceMap) - 1
5
6 }
```

```
1 ------------------ RUST TRANSLATION ------------------
2 pub(crate) fn add_sentence(ranks: &mut Rank,
↪    sentence: ParsedSentence) -> Result<i32, Error> {
3     let sentence_id = ranks.sentence_map.len() as i32;
4     ranks.sentence_map.insert(sentence_id,
↪        sentence.original.clone());
5     Ok(sentence_id)
6 }
```

The next code snippet demonstrates an example of an **Excessively Strict Equivalence Definition** taken from the `deltachat-core` project. Both the C and Rust function retrieve a field `blobdir`. MATCHFIXAGENT states that these are not equivalent because (1) the Rust function does not perform a null check, and (2) the C function returns a copy of `blobdir` whereas the Rust returns a reference. While this is true, the translation follows Rust's idioms, and a developer would not care about these differences. Rust's type system prevents `blobdir` from ever being null, and the Rust translation returns an *immutable* reference, preventing the caller from modifying the return value.

```
1 ------------------ C SOURCE CODE ------------------
2 char* dc_get_blobdir(const dc_context_t* context) {
3     if (context==NULL ||
↪        context->magic!=DC_CONTEXT_MAGIC) {
4         return dc_strdup(NULL);
5     }
6     return dc_strdup(context->blobdir);
7 }
```

```
1 ------------------ RUST TRANSLATION ------------------
2 pub fn get_blobdir(&self) -> &Path {
3
4
5     &self.inner.blobdir
6
7
8
9 }
```

### D.3. Analysis of the Agreements

From the original experiments using `Claude 3.7 Sonnet`, we uniformly sampled at most 5 equivalent validations across 24 benchmarks and four validation systems, resulting in 110 source–target pairs. In total, 88 were correctly validated as

equivalent, and 22 validated as non-equivalent by both existing tools and MATCHFIXAGENT. The same two authors from dispute analysis (§D.1) performed manual investigation of equivalent cases with $95.5\%$ inter-rater agreement and show that $\frac{106}{110}$ ($96.4\%$) judgements were correct, meaning $\frac{4}{110}$ ($3.6\%$) were incorrect validations, indicating a need for more rigorous testing and oversight in MATCHFIXAGENT.

The following code snippets show an example of false positive, where both MATCHFIXAGENT and OXIDIZER validate the translation as equivalent, but the human investigator concludes otherwise. The translated Rust code incorrectly returns an error if any word is not found in `word_val_id`, while the Go source code silently handles missing words and returns zero for missing keys and continues. This difference in error handling means they behave differently when words are not in the map. OXIDIZER and MATCHFIXAGENT failed to detect this bug due to insufficient testing.

```
1  ------------------ GO SOURCE CODE ------------------
2  func FindSentencesByPhrases(ranks *Rank, words
   ↪  []string) []Sentence {
3
4
5
6      for _, i := range words {
7          for _, j := range words {
8              x := ranks.WordValID[i]
9              y := ranks.WordValID[j]
10             ...
11         }
12     }
13
14
15
16     ...
17     return sentences
18 }
```

```
1  ----------------- RUST TRANSLATION -----------------
2  pub fn find_sentences_by_phrases(ranks:
   ↪  Option<&Rank>, words: &[String]) ->
   ↪  Result<Vec<Sentence>> {
3
4      for i in words {
5          for j in words {
6              let x = *ranks.word_val_id.get(i)
7                  .ok_or_else( anyhow::
8                  anyhow!("Word not found: {}", i))?;
9              let y = *ranks.word_val_id.get(j)
10                 .ok_or_else( anyhow::
11                 anyhow!("Word not found: {}", j))?;
12             ...
13         }
14     }
15     ...
16     Ok(sentences)
17 }
```

The next code snippets demonstrate an example of false negative, where both MATCHFIXAGENT and RUSTREPOTRANS validate the translation as non-equivalent, but the human investigator concludes otherwise. Both functions compute the current time in slots since the Unix epoch by dividing the current Unix timestamp (in seconds) by `(60 * TIME_SLOT_MINUTES)`. The logic is identical, with only language-specific differences in how current time is obtained and types used (`unsign32` vs `usize`). RUSTREPOTRANS and MATCHFIXAGENT validate these as functionally inequivalent because there is a compilation bug somewhere else in the codebase.

```
1  ------------------ C SOURCE CODE ------------------
2      unsign32 today(void)
3      {
4          /* return time in slots since epoch */
5          unsign32 ti = (unsign32) time(NULL);
6          return ti / (60 * TIME_SLOT_MINUTES);
7      }
```

```
1  ----------------- RUST TRANSLATION -----------------
2      pub fn today() -> usize {
3          // Return time in slots since epoch
4          let now = SystemTime::now().duration_since(UN ↲
            ↪  IX_EPOCH).unwrap().as_secs() as usize;
5          now / (60 * rom::TIME_SLOT_MINUTES)
6      }
```

## E. Threshold Ablation

Table 5 shows the results of our ablation study on the threshold $\tau$ in Algorithm 2 and Algorithm 3. The threshold $\tau$ governs a tradeoff between algorithmic reliance and LLM invocation cost in the Control Flow and Data Flow Path analyzers, which are the only two sub-analyzers with a short-circuit mechanism; the remaining four (I/O Mapping, Library API, Exception Handling, Specifications) are unaffected and serve as an internal control. A lower $\tau$ is easier to satisfy, causing the short-circuit to fire more frequently and reducing LLM cost, but over-trusting the structural similarity algorithm: at $\tau=0.5$, TNR on Control Flow and Data Flow Path drops to $19.7\%$ and $17.7\%$ respectively—meaning the structural algorithm alone fails to correctly identify inequivalent pairs in roughly four out of five cases. A higher $\tau$ is harder to satisfy, so fewer cases are short-circuited and more LLM invocations are triggered, increasing cost. Moving from $\tau=0.5$ to $\tau=0.7$ yields the largest correctness gains: TNR on Control Flow jumps $+14.6\%$ at a TPR cost of only $-2.6\%$, and TNR on Data Flow Path gains $+5.6\%$ at a negligible $-0.2\%$ TPR cost. Moving further to $\tau=0.9$ yields only marginal additional TNR improvements ($+8.4\%$ and $+3.5\%$ respectively) while TPR continues to erode ($-1.6\%$ and $-1.4\%$) and LLM invocation cost increases. The average TNR across all six analyzers follows the same pattern: $34.3\%$ at $\tau=0.5$, $37.8\%$ at $\tau=0.7$, and $39.8\%$ at $\tau=0.9$, confirming that $\tau=0.7$ captures the steepest part of the improvement curve. We therefore select $\tau=0.7$ as the main threshold value—delivering substantial TNR improvement over $\tau=0.5$ without the diminishing returns and added cost of $\tau=0.9$. Nonetheless, since the threshold functions as a cost-optimization lever rather than a correctness lever—with any errors in a single sub-analyzer recoverable by the downstream Test Generator & Repair Agent and Verdict Agent—practitioners can adjust $\tau$ based on their own risk tolerance and computational budget, using the TPR/TNR/FPR/FNR results in Table 5.

*Table 5.* Agreement between each semantic analyzer and MATCHFIXAGENT verdict. Experiments repeated with the Claude model for threshold $\tau \in \{0.5, 0.7, 0.9\}$. **TP:** True Positive (Semantic Analyzer=Yes and MATCHFIXAGENT =Yes), **FP:** False Positive (Semantic Analyzer=Yes and MATCHFIXAGENT =No), **FN:** False Negative (Semantic Analyzer=No and MATCHFIXAGENT =Yes), **TN:** True Negative (Semantic Analyzer=No and MATCHFIXAGENT =No).

| Semantic Analyzer | $\tau$ | TP | | FP | | FN | | TN | |
|---|---|---|---|---|---|---|---|---|---|
| | | Count | TPR | Count | FPR | Count | FNR | Count | TNR |
| Control Flow | 0.5 | 1396 | 0.957 | 549 | 0.803 | 63 | 0.043 | 135 | 0.197 |
| | 0.7 | 1404 | 0.931 | 438 | 0.657 | 104 | 0.069 | 229 | 0.343 |
| | 0.9 | 1342 | 0.915 | 375 | 0.573 | 124 | 0.085 | 279 | 0.427 |
| Data Flow Path | 0.5 | 1411 | 0.968 | 552 | 0.823 | 47 | 0.032 | 119 | 0.177 |
| | 0.7 | 1459 | 0.966 | 510 | 0.767 | 52 | 0.034 | 155 | 0.233 |
| | 0.9 | 1395 | 0.952 | 483 | 0.732 | 70 | 0.048 | 177 | 0.268 |
| Input/Output Mapping | 0.5 | 1399 | 0.961 | 323 | 0.474 | 57 | 0.039 | 358 | 0.526 |
| | 0.7 | 1437 | 0.953 | 321 | 0.481 | 71 | 0.047 | 347 | 0.519 |
| | 0.9 | 1416 | 0.963 | 322 | 0.481 | 55 | 0.037 | 348 | 0.519 |
| Library API Equivalence | 0.5 | 1441 | 0.992 | 478 | 0.714 | 12 | 0.008 | 191 | 0.286 |
| | 0.7 | 1481 | 0.988 | 482 | 0.730 | 18 | 0.012 | 178 | 0.270 |
| | 0.9 | 1450 | 0.989 | 477 | 0.726 | 16 | 0.011 | 180 | 0.274 |
| Exception/Error Handling | 0.5 | 1396 | 0.960 | 433 | 0.640 | 58 | 0.040 | 244 | 0.360 |
| | 0.7 | 1420 | 0.940 | 425 | 0.636 | 91 | 0.060 | 243 | 0.364 |
| | 0.9 | 1407 | 0.959 | 428 | 0.641 | 60 | 0.041 | 240 | 0.359 |
| Specifications | 0.5 | 1378 | 0.953 | 329 | 0.486 | 68 | 0.047 | 348 | 0.514 |
| | 0.7 | 1406 | 0.938 | 308 | 0.462 | 93 | 0.062 | 359 | 0.538 |
| | 0.9 | 1376 | 0.947 | 308 | 0.460 | 77 | 0.053 | 362 | 0.540 |
| Average | 0.5 | 1403 | 0.965 | 444 | 0.657 | 50 | 0.035 | 232 | 0.343 |
| | 0.7 | 1434 | 0.953 | 414 | 0.622 | 71 | 0.047 | 251 | 0.378 |
| | 0.9 | 1397 | 0.954 | 398 | 0.602 | 67 | 0.046 | 264 | 0.398 |

## F. MATCHFIXAGENT with Open Source Model

MATCHFIXAGENT is LLM-agnostic and easily adapts to new models. To show its open-source adaptability, we reproduced Table 1 with the state-of-the-art `Qwen3-Next-80B-A3B` (released February 2026). Table 6 shows the results of this experiment, indicating that the agreement rate of Qwen3 (71.6%) remains consistent with Claude (72.8%), suggesting similarity between the open-source and closed LLMs.

*Table 6.* Effectiveness of MATCHFIXAGENT in translation validation compared to existing techniques using `Qwen3-Next-80B-A3B`. **EQ:** Equivalent, **NEQ:** Not Equivalent, **VF:** Validation Failure, **Agreement**: number and percentage of translation pairs where MATCHFIX-AGENT's and the existing tool's verdicts agree, **Disagreement**: percentage of disagreements ruled in favor of **Tool** and MATCHFIXAGENT (**Ours**). **VF**s are excluded from **Agreement** and **Disagreement** calculations.

| Tool | Project | Total # Trans. Pairs | Tool Validation | | | MATCHFIXAGENT | | | Agreement |
|---|---|---|---|---|---|---|---|---|---|
| | | | EQ | NEQ | VF | EQ | NEQ | VF | |
| OXIDIZER | checkdigit | 29 | 21 (72.4) | 8 (27.6) | 0 (0) | 23 (79.3) | 6 (20.7) | 0 (0) | 19 (65.5) |
| | go-edlib | 24 | 18 (75) | 6 (25) | 0 (0) | 14 (58.3) | 10 (41.7) | 0 (0) | 12 (50) |
| | histogram | 19 | 12 (63.2) | 7 (36.8) | 0 (0) | 14 (73.7) | 5 (26.3) | 0 (0) | 7 (36.8) |
| | nameparts | 15 | 9 (60) | 6 (40) | 0 (0) | 10 (66.7) | 5 (33.3) | 0 (0) | 14 (93.3) |
| | stats | 53 | 38 (71.7) | 14 (26.4) | 1 (1.9) | 35 (66) | 17 (32.1) | 1 (1.9) | 33 (64.7) |
| | textrank | 52 | 40 (76.9) | 12 (23.1) | 0 (0) | 32 (61.5) | 20 (38.5) | 0 (0) | 32 (61.5) |
| **Total** | | **192** | **138 (71.9)** | **53 (27.6)** | **1 (0.5)** | **128 (66.7)** | **63 (32.8)** | **1 (0.5)** | **117 (61.6)** |
| ALPHA TRANS | cli | 273 | 210 (76.9) | 24 (8.8) | 39 (14.3) | 214 (78.4) | 53 (19.4) | 6 (2.2) | 173 (75.2) |
| | csv | 235 | 97 (41.3) | 61 (26) | 77 (32.8) | 155 (66) | 75 (31.9) | 5 (2.1) | 93 (60) |
| | fileupload | 192 | 19 (9.9) | 1 (0.5) | 172 (89.6) | 143 (74.5) | 43 (22.4) | 6 (3.1) | 17 (85) |
| | validator | 646 | 247 (38.2) | 103 (15.9) | 296 (45.8) | 475 (73.5) | 158 (24.5) | 13 (2) | 223 (65) |
| **Total** | | **1346** | **573 (42.6)** | **189 (14)** | **584 (43.4)** | **987 (73.3)** | **329 (24.4)** | **30 (2.2)** | **506 (67.6)** |
| SKEL | bst | 19 | 19 (100) | 0 (0) | 0 (0) | 13 (68.4) | 5 (26.3) | 1 (5.3) | 13 (72.2) |
| | colorsys | 8 | 8 (100) | 0 (0) | 0 (0) | 8 (100) | 0 (0) | 0 (0) | 8 (100) |
| | heapq | 22 | 19 (86.4) | 3 (13.6) | 0 (0) | 19 (86.4) | 3 (13.6) | 0 (0) | 20 (90.9) |
| | html | 44 | 40 (90.9) | 2 (4.5) | 2 (4.5) | 23 (52.3) | 19 (43.2) | 2 (4.5) | 24 (60) |
| | mathgen | 81 | 77 (95.1) | 4 (4.9) | 0 (0) | 66 (81.5) | 12 (14.8) | 3 (3.7) | 66 (84.6) |
| | rbt | 27 | 26 (96.3) | 0 (0) | 1 (3.7) | 19 (70.4) | 5 (18.5) | 3 (11.1) | 18 (78.3) |
| | strsim | 64 | 50 (78.1) | 0 (0) | 14 (21.9) | 56 (87.5) | 6 (9.4) | 2 (3.1) | 45 (91.8) |
| | toml | 72 | 37 (51.4) | 10 (13.9) | 25 (34.7) | 46 (63.9) | 23 (31.9) | 3 (4.2) | 35 (77.8) |
| **Total** | | **337** | **276 (81.9)** | **19 (5.6)** | **42 (12.5)** | **250 (74.2)** | **73 (21.7)** | **14 (4.2)** | **229 (80.9)** |
| RUSTREPO TRANS | charset | 33 | 20 (60.6) | 13 (39.4) | 0 (0) | 11 (33.3) | 21 (63.6) | 1 (3) | 20 (62.5) |
| | deltachat | 125 | 54 (43.2) | 69 (55.2) | 2 (1.6) | 30 (24) | 91 (72.8) | 4 (3.2) | 87 (73.1) |
| | iceberg-java | 25 | 9 (36) | 16 (64) | 0 (0) | 4 (16) | 20 (80) | 1 (4) | 20 (83.3) |
| | iceberg-py | 44 | 15 (34.1) | 28 (63.6) | 1 (2.3) | 6 (13.6) | 36 (81.8) | 2 (4.5) | 34 (82.9) |
| | crypto-c | 20 | 16 (80) | 4 (20) | 0 (0) | 10 (50) | 7 (35) | 3 (15) | 14 (82.4) |
| | crypto-java | 97 | 39 (40.2) | 58 (59.8) | 0 (0) | 26 (26.8) | 68 (70.1) | 3 (3.1) | 82 (87.2) |
| **Total** | | **344** | **153 (44.5)** | **188 (54.7)** | **3 (0.9)** | **87 (25.3)** | **243 (70.6)** | **14 (4.1)** | **257 (78.6)** |
| **Total** | | **2219** | **1140 (51.4)** | **449 (20.2)** | **630 (28.4)** | **1452 (65.4)** | **708 (31.9)** | **59 (2.7)** | **1109 (71.6)** |

## G. Cost Analysis

Table 7 shows the cost analysis of different components in MATCHFIXAGENT for two models, namely, `Claude 3.7 Sonnet` and `Qwen3-Next-80B-A3B`. The results indicate the open-source Qwen3 model is on average $\times 23.8$ cheaper than Claude. Please note that we rely on the numbers reported by Claude Code for input and output tokens, and calculate dollar ($) cost accordingly. For instance Claude requires $3 / 1 million input tokens and $15 / 1 million output tokens, and Qwen3 requires $0.09 / 1 million input tokens and $0.78 / 1 million output tokens. Since we run our Qwen3 experiments using a proxy server (e.g., LiteLLM (Team, 2026c)) for compatibility purposes, the number of cached input tokens were not reported properly and as a result there is a significant incorrect difference, for instance, $2M$ compared to 166 average input tokens for Test Generator and Repair agent across all tools. However, after checking the logs from the LLM provider, we noticed that the cost was correctly reflected, in the same order of magnitude, for the open-source Qwen3 model.

## H. Optimizing MATCHFIXAGENT

There are many directions that can be explored to optimize both the dollar cost and the runtime of MATCHFIXAGENT, which we outline below, but leave this as future work. However, please note though that MATCHFIXAGENT 's current dollar cost and runtime compare favorably to the development cost of existing repository-level validation techniques, or even when compared to a salaried human software engineer manually performing validation. Compared to these two baselines, 309s and $1.22 per translation pair are not daunting. A fully-automated translation tool that runs for a week and costs a few $100 is still extremely useful in practice if the translation produced is high quality. The following are potential optimizations:

1. Multiple instances of MATCHFIXAGENT can run concurrently on functions that are not dependent on each other. We

*Table 7.* Cost analysis of MATCHFIXAGENT components. Tuple entries indicate ⟨`Claude 3.7 Sonnet`,`Qwen3-Next-80B-A3B`⟩.

| Tool | Total # Trans. Pairs | Component | Avg Input Tokens | Avg Output Tokens | Avg Duration (ms) | Avg Cost ($) |
|---|---|---|---|---|---|---|
| ALPHATRANS | 1346 | Control Flow | ⟨437.0, 349.5⟩ | ⟨86.9, 65.4⟩ | ⟨2557.5, 2557.5⟩ | ⟨0.00261, 0.00008⟩ |
| | | Data Flow | ⟨1124.5, 931.8⟩ | ⟨122.8, 93.9⟩ | ⟨3507.8, 3507.8⟩ | ⟨0.00522, 0.00016⟩ |
| | | IO | ⟨782.4, 646.0⟩ | ⟨205.0, 217.5⟩ | ⟨7059.7, 7059.7⟩ | ⟨0.00542, 0.00023⟩ |
| | | Lib Equivalence | ⟨700.4, 583.0⟩ | ⟨235.9, 218.2⟩ | ⟨6900.2, 6900.2⟩ | ⟨0.00564, 0.00022⟩ |
| | | Exception Error | ⟨726.4, 599.0⟩ | ⟨152.0, 182.2⟩ | ⟨6250.3, 6250.3⟩ | ⟨0.00446, 0.00020⟩ |
| | | Spec | ⟨726.4, 599.0⟩ | ⟨311.1, 318.6⟩ | ⟨8801.8, 8801.8⟩ | ⟨0.00685, 0.00030⟩ |
| | | Test Gen & Repair | ⟨161.6, 2104327.0⟩ | ⟨11527.8, 7672.8⟩ | ⟨274732.6, 294980.7⟩ | ⟨0.17340, 0.19537⟩ |
| | | Verdict | ⟨36.6, 32582.1⟩ | ⟨1404.6, 240.2⟩ | ⟨31737.7, 9770.2⟩ | ⟨0.02118, 0.00312⟩ |
| OXIDIZER | 192 | Control Flow | ⟨1426.4, 1186.9⟩ | ⟨230.8, 168.5⟩ | ⟨3957.4, 3957.4⟩ | ⟨0.00774, 0.00024⟩ |
| | | Data Flow | ⟨2298.7, 1928.8⟩ | ⟨206.6, 159.1⟩ | ⟨3726.5, 3726.5⟩ | ⟨0.00999, 0.00030⟩ |
| | | IO | ⟨993.8, 833.6⟩ | ⟨238.7, 297.3⟩ | ⟨6532.9, 6532.9⟩ | ⟨0.00656, 0.00031⟩ |
| | | Lib Equivalence | ⟨912.8, 770.6⟩ | ⟨265.7, 250.3⟩ | ⟨5640.9, 5640.9⟩ | ⟨0.00672, 0.00026⟩ |
| | | Exception Error | ⟨937.8, 786.6⟩ | ⟨248.3, 240.8⟩ | ⟨6100.1, 6100.1⟩ | ⟨0.00654, 0.00026⟩ |
| | | Spec | ⟨937.8, 786.6⟩ | ⟨399.7, 408.7⟩ | ⟨10097.8, 10097.8⟩ | ⟨0.00881, 0.00039⟩ |
| | | Test Gen & Repair | ⟨165.2, 2415982.5⟩ | ⟨11340.1, 8769.0⟩ | ⟨244057.2, 281051.7⟩ | ⟨0.17060, 0.22428⟩ |
| | | Verdict | ⟨43.1, 28828.1⟩ | ⟨1813.4, 264.8⟩ | ⟨37238.3, 6736.6⟩ | ⟨0.02733, 0.00280⟩ |
| RUSTREPOTRANS | 344 | Control Flow | ⟨2247.7, 1757.5⟩ | ⟨266.4, 187.0⟩ | ⟨5437.8, 5437.8⟩ | ⟨0.01074, 0.00030⟩ |
| | | Data Flow | ⟨5083.8, 4841.3⟩ | ⟨275.7, 186.1⟩ | ⟨4229.3, 4229.3⟩ | ⟨0.01939, 0.00058⟩ |
| | | IO | ⟨1167.0, 939.1⟩ | ⟨279.0, 307.9⟩ | ⟨7104.9, 7104.9⟩ | ⟨0.00769, 0.00032⟩ |
| | | Lib Equivalence | ⟨1086.0, 876.1⟩ | ⟨318.4, 275.4⟩ | ⟨6532.2, 6532.2⟩ | ⟨0.00803, 0.00029⟩ |
| | | Exception Error | ⟨1111.0, 892.1⟩ | ⟨287.8, 253.5⟩ | ⟨6326.0, 6326.0⟩ | ⟨0.00765, 0.00028⟩ |
| | | Spec | ⟨1111.0, 889.8⟩ | ⟨491.5, 457.2⟩ | ⟨9488.8, 9488.8⟩ | ⟨0.01071, 0.00044⟩ |
| | | Test Gen & Repair | ⟨189.7, 1329259.2⟩ | ⟨15020.1, 4355.7⟩ | ⟨401964.1, 148608.3⟩ | ⟨0.22587, 0.12303⟩ |
| | | Verdict | ⟨44.3, 84397.6⟩ | ⟨1925.5, 327.8⟩ | ⟨44992.7, 10575.6⟩ | ⟨0.02902, 0.00785⟩ |
| SKEL | 337 | Control Flow | ⟨1163.1, 949.7⟩ | ⟨201.4, 153.5⟩ | ⟨3640.0, 3640.0⟩ | ⟨0.00651, 0.00021⟩ |
| | | Data Flow | ⟨3621.2, 4951.7⟩ | ⟨258.6, 195.3⟩ | ⟨4867.5, 4867.5⟩ | ⟨0.01474, 0.00060⟩ |
| | | IO | ⟨907.7, 761.1⟩ | ⟨218.5, 245.6⟩ | ⟨5834.6, 5834.6⟩ | ⟨0.00600, 0.00026⟩ |
| | | Lib Equivalence | ⟨826.7, 698.1⟩ | ⟨233.5, 221.2⟩ | ⟨5237.1, 5237.1⟩ | ⟨0.00598, 0.00024⟩ |
| | | Exception Error | ⟨851.7, 714.1⟩ | ⟨157.5, 192.7⟩ | ⟨4896.8, 4896.8⟩ | ⟨0.00492, 0.00021⟩ |
| | | Spec | ⟨851.7, 714.1⟩ | ⟨368.4, 378.8⟩ | ⟨8539.1, 8539.1⟩ | ⟨0.00808, 0.00036⟩ |
| | | Test Gen & Repair | ⟨160.8, 1843194.0⟩ | ⟨11803.7, 8944.5⟩ | ⟨245430.8, 272173.9⟩ | ⟨0.17754, 0.17286⟩ |
| | | Verdict | ⟨38.8, 57464.5⟩ | ⟨1421.1, 355.8⟩ | ⟨37516.6, 9687.9⟩ | ⟨0.02143, 0.00545⟩ |
| Average | 2219 | Control Flow | ⟨913.6, 731.4⟩ | ⟨144.6, 106.6⟩ | ⟨3289.5, 3289.5⟩ | ⟨0.00491, 0.00015⟩ |
| | | Data Flow | ⟨2219.1, 2234.6⟩ | ⟨174.4, 129.2⟩ | ⟨3845.1, 3845.1⟩ | ⟨0.00927, 0.00030⟩ |
| | | IO | ⟨879.3, 725.1⟩ | ⟨221.4, 242.7⟩ | ⟨6835.0, 6835.0⟩ | ⟨0.00596, 0.00025⟩ |
| | | Lib Equivalence | ⟨797.7, 662.1⟩ | ⟨250.9, 230.3⟩ | ⟨6481.6, 6481.6⟩ | ⟨0.00616, 0.00024⟩ |
| | | Exception Error | ⟨823.3, 678.1⟩ | ⟨182.2, 199.9⟩ | ⟨6043.5, 6043.5⟩ | ⟨0.00520, 0.00022⟩ |
| | | Spec | ⟨823.3, 677.8⟩ | ⟨355.4, 357.0⟩ | ⟨8980.5, 8980.5⟩ | ⟨0.00780, 0.00034⟩ |
| | | Test Gen & Repair | ⟨166.1, 1971480.1⟩ | ⟨12094.9, 7446.6⟩ | ⟨287352.3, 267620.4⟩ | ⟨0.18192, 0.18324⟩ |
| | | Verdict | ⟨38.7, 44068.9⟩ | ⟨1523.3, 273.4⟩ | ⟨35146.1, 9620.0⟩ | ⟨0.02296, 0.00418⟩ |

can determine function dependencies by constructing a call graph, which is a common program analysis technique. This approach would reduce the total runtime and does not require any changes to the design of MATCHFIXAGENT.

2. Use faster, cheaper LLMs or skip analysis for "trivial" translation pairs (e.g., simple `getter` / `setter` functions or those using only primitive types). Heuristics could detect these to avoid expensive semantic analysis or use a cheaper LLM.

3. A single instance of MATCHFIXAGENT can process multiple related functions at once. Part of the overhead of running a coding agent is when the agent "orients" itself by reading code files to understand the codebase. By processing multiple related functions at once, we eliminate redundant "orientation" phases, reducing both runtime and cost.

## I. Threats to Validity

Similar to prior techniques, MATCHFIXAGENT comes with some limitations and threats to the validity. In this section, we discuss how we mitigated various threats.

**Internal Validity.** There are two main threats to internal validity. First, we only run experiments once. Since LLMs are inherently non-deterministic, running experiments again may produce different results. While it is highly *likely* some individual equivalence verdicts and repair results would change if experiments were run again, it is highly *unlikely* the aggregate metrics we report would change significantly given the large number of translation samples we use (2,219 pairs). Second, our human investigation does not assess ground truth equivalence. We only assess whether an inequivalent verdict was correct, but we do not analyze the correctness of equivalent verdicts. While this means we don't have any measure of true accuracy of MATCHFIXAGENT, we still can claim MATCHFIXAGENT is more accurate than existing automated validation techniques.

**External Validity.** One main external threat is the generalizability of our approach. Our validation and repair system is very generic and can be extended to more PL pairs with minimal engineering effort. Also, the majority of tools that we used, for example, Tree-Sitter (Tree-Sitter, 2026) can support a large set of PLs. To mitigate external validity, we built the initial version of MATCHFIXAGENT with six PLs.

**Construct Validity.** In order to minimize construct validity, MATCHFIXAGENT is built on well-known and rigorously tested tools, e.g., Tree-Sitter (Tree-Sitter, 2026), Claude Code (Anthropic, 2026c), and Codex (OpenAI, 2026a).

## J. Figures

```
============= Source Code =============
def max(a, b):
    """
    The function max(a, b) returns the
    greater of the two input values a
    and b.  If a is greater than b, it
    returns a; otherwise, it returns a.
    """
    if a > b:
        return a
    else:
        return b
```

```
========== Control Flow Graph =========
Node 0:  PARAMETERS
  parameter:  a, parameter:  b
  → Node 1:  ENTRY
Node 1:  ENTRY
  → Node 2:  if a > b:
Node 2:
  if a > b:
  → Node 3 [label="a > b"]:
    return a
  → Node 4 [label="(a <= b)"]:
    return b
```

```
========== Data Flow Paths ===========
Variable:  a
  Path 1:
    CFG Node Path:  0 → 1 → 2
  Path 2:
    CFG Node Path:  0 → 1 → 2 → 3

Variable:  b
  Path 1:
    CFG Node Path:  0 → 1 → 2
  Path 2:
    CFG Node Path:  0 → 1 → 2 → 4
```

*Figure 6.* CFG and DFP structures extracted by the Semantic Analyzer component in MATCHFIXAGENT.

```
<fragment_details>
  <source_fragment_details> <path to source file> <implementation of source fragment> </source_fragment_details>
  <target_fragment_details> <path to target file> <implementation of target fragment> </target_fragment_details>
</fragment_details>

<instruction>
  You are an expert agent specializing in test generation and code repair.  Based on the analysis from multiple expert agents
  regarding functional equivalence between $SOURCE_LANGUAGE and $TARGET_LANGUAGE implementations of the given method/function, your
  task is to generate tests and repair the target implementation, if necessary.
  <functional_equivalence_definition>
    Two code fragments in different programming languages are considered functionally equivalent if, when executed on the same
    input, they always have identical program states at all corresponding points reachable by program execution, and they both
    produce the same output upon termination.
  </functional_equivalence_definition>
  <rules_and_general_notes> 1.  Consider the Semantic Analyzer results ..., 2.  Generate tests ...  </rules_and_general_notes>
</instruction>

<semantic_analysis_results> {"control_flow": <>, "data_flow": <>, "io": <>, "lib_api": <>, ...} </semantic_analysis_results>

<final_response_format> {"is_equivalent": <>, "explanation": <>, "tests": <>, "patch": <>, ...} </final_response_format>
```

*Figure 7.* Prompt structure of the Test Generator and Repair Agent.

