# OpenReview forum: "MatchFixAgent: Language-Agnostic Autonomous Repository-Level Code Translation Validation and Repair"
_ICML.cc/2026/Conference — ICML 2026 regular_

### Official Review · Reviewer_f59W · 2026-03-06

**Soundness:** 2
**Presentation:** 3
**Significance:** 2
**Originality:** 3
**Overall Recommendation:** 3
**Confidence:** 4

**Summary:**

This paper proposes MatchFixAgent, an LLM-based, language-agnostic multi-agent framework designed for the validation and repair of repository-level code translations. The system divides the validation process into several sub-tasks, such as control flow, data flow, and library API analysis, to conduct a comprehensive semantic analysis, and utilizes a test generator and repair agent to handle discrepancies.

**Compliance With Llm Reviewing Policy:**

Affirmed.

**Final Justification:**

The authors have made a commendable effort during the rebuttal to clarify their design trade-offs. However, because the system relies on syntactic approximations that run the risk of letting subtle bugs slip into production, the paper requires revisions before it can be meaningfully built upon by others. As such, my recommendation remains 3.

**Key Questions For Authors:**

1. Can you provide sufficient scientific justification or ablation data to prove that setting the threshold to 0.7 in the structural similarity calculations is reasonable?

2. Given that the paper currently have‘nt evaluate the ground truth of "equivalent" verdicts, how do you plan to address this issue?

3. Why was the ablation study conducted exclusively on "non-dispute" samples? How does this design prove the true value of the multi-agent architecture in handling complex or difficult code translations?

4. How will the authors optimize this system to make it truly viable for validating large code repositories in the real world?

**Limitations:**

If a developer blindly trusts a false "equivalent" verdict produced by the tool, it could lead to subtle logic bugs or masked security vulnerabilities being unknowingly pushed into production environments.

**Strengths And Weaknesses:**

Despite its attempts at cross-language compatibility, the framework exhibits significant weaknesses in the depth of its underlying analysis, the rigor of its experimental evaluation, and its practical deployment costs.

1. The paper claims to perform data flow analysis, but this relies primarily on syntactic analysis using Tree-sitter. The authors explicitly state that this method does not handle complex program analysis challenges such as aliasing, concurrency, or context sensitivity, undermining the reliability of the analysis when applied to complex, enterprise-level code in the real world.

2. In both the control flow and data flow analysis modules, the authors hardcode a similarity threshold of 0.7 to bypass LLM invocations. The paper provides no ablation studies or theoretical justification for why 0.7 is the optimal threshold, only noting that it skips approximately 25% and 35% of LLM invocations, respectively.

3. The evaluation lacks a baseline to verify whether the code is genuinely "functionally equivalent". The authors admit that their manual investigation only assessed the correctness of the tool when it returned a "not equivalent" verdict, but did not analyze the correctness of "equivalent" verdicts. They also acknowledge that this means they do not have any measure of the true accuracy of MatchFixAgent.

4. Although the system is advertised for "repository-level" code translation, its operating costs are daunting. According to the report, producing a verdict takes an average of 309 seconds per translation pair, costs $1.22 per pair, and totaled $2,710.45 for the experiments.

---

> ### Author Rebuttal · Authors · 2026-03-31
>
> We would like to thank the reviewer for providing constructive feedback. Please find our response to your questions below. Please feel free to ask additional clarifying questions.
>
> **Q1. Can you provide sufficient scientific justification or ablation data to prove that setting the threshold to 0.7 in the structural similarity calculations is reasonable?**
>
> We acknowledge that 0.7 is a hyperparameter of MatchFixAgent, chosen as a reasonably conservative operating point on the efficiency-thoroughness tradeoff. Crucially, this threshold functions purely as a cost-optimization lever, not a correctness lever: it only determines whether an LLM call is skipped for a single sub-analyzer, and any resulting error is recoverable by the downstream Test Generator & Repair Agent and Verdict Agent, which independently assess equivalence through test execution. Furthermore, the threshold can only trigger in one direction—short-circuiting on high structural similarity—bounding its failure mode to cases already likely to be equivalent. The moderate empirical skip rates it produces (25% for CFG, 35% for DFG) confirm it is not overly aggressive, and the system's overall accuracy—judged correct in 60.7% of disagreements with prior work—shows no evidence of threshold-induced systematic errors. We therefore argue that while 0.7 is a hyperparameter, the multi-agent architecture renders the system naturally robust to reasonable variation in its value. A user of MatchFixAgent can adjust this threshold based on their tradeoffs and risk factors.
>
> **Q2. Given that the paper currently hasn't evaluated the ground truth of "equivalent" verdicts, how do you plan to address this issue?**
>
> Please see our response to **Reviewer u28r Q1** as they ask the same question.
>
> **Q3. Why was the ablation study conducted exclusively on "non-dispute" samples?**
>
> The non-dispute samples were chosen for the ablation specifically to ensure methodological soundness: on disputed samples, ground truth is unknown without expensive manual investigation, making it impossible to reliably measure the baseline agent's accuracy. Non-dispute samples, where MatchFixAgent and prior tools independently converge on the same verdict, provide a clean and unambiguous reference signal for controlled comparison. Essentially, this design choice is actually conservative—non-dispute samples are arguably the easier subset, yet the standalone agent still suffers a 42.3% accuracy drop, suggesting the true degradation on harder disputed cases would likely be even larger. We acknowledge that an ablation directly on disputed samples is a valid and interesting direction—it would require additional manual ground truth labeling of those cases, which we consider worthwhile future work and will note explicitly in the paper.
>
> **Q4. How will the authors optimize this system to make it truly viable for validating large code repositories in the real world?**
>
> While we acknowledge the challenge in validating repository-level code, we offer a comparison based on Tool Lines of Code (LoC) and Tool Development Time (months), confirmed by the authors of existing techniques. Each of these techniques only supports one language pair (e.g., Oxidizer: Go-Rust). Although existing tools may have cheaper runtime costs for validating a single translation pair, developing them across multiple language pairs remains challenging. Thus, MatchFixAgent offers a more scalable solution for multiple languages.
>
> | Technique | Tool LoC | Development Time (Months) |
> | :--- | :--- | :--- |
> | **Oxidizer** | 19,052 | 10 |
> | **AlphaTrans** | 10,859 | 8 |
> | **Skel** | 3,843 | 6 |
> | **MatchFixAgent** | 1,650 | 0.7 |
>
> **===== Additional Concerns =====**
>
> **C1. The paper claims to perform data flow analysis, but this relies primarily on syntactic analysis using Tree-sitter.**
>
> We explicitly acknowledge this limitation in the paper (Lines 166–169), but agree it deserves more prominent framing and will update the terminology accordingly. This design is however deliberate and justified by MatchFixAgent’s core philosophy: full semantic data flow analysis requires language-specific compilers, intermediate representations, and deep engineering effort—precisely the overhead that restricts prior tools to a single language pair. Our syntactic approximation requires only ~280 lines of code per language while still capturing meaningful structural features about variable propagation. Essentially, the syntactic extraction and the LLM are complementary by design—the def-use chains provide structured variable flow information that the LLM uses as scaffolding for deeper semantic reasoning it would otherwise have to infer from raw code alone. The ablation study corroborates this: removing the semantic analyzer including the data flow component causes a 42.3% accuracy drop, confirming that even this syntactic approximation contributes meaningfully to system performance.

---

> > ### Author Rebuttal · Reviewer_f59W · 2026-04-03
> >
> > Thank you for the detailed clarifications; however, I would appreciate it if you could provide the specific ablation data justifying the 0.7 threshold as initially requested, and explain why the ablation study was exclusively conducted on non-dispute samples. Additionally, while adopting open-source models addresses some cost concerns, further insights into how you plan to structurally optimize the system to handle the high operational costs for real-world repositories would be very helpful.

---

> > > ### Author Response · Authors · 2026-04-07
> > >
> > > **Q1. Threshold Ablation**
> > >
> > > We re-ran all experiments with new thresholds (t values 0.5 and 0.9) to produce TPR (DFG/CFG=Yes, MatchFixAgent=Yes) and TNR (DFG/CFG=No, MatchFixAgent=No) values, consistent with our response to **Reviewer d9RK, Q5**.
> > >
> > > | Analyzer       | Metric | t = 0.5 | t = 0.7 | t = 0.9 |
> > > |----------------|--------|---------|---------|---------|
> > > | Control Flow   | TPR    | 95.7%   | 93.1%   | 91.4%   |
> > > | Control Flow   | TNR    | 19.9%   | 34.3%   | 43.0%   |
> > > | Data Flow | TPR    | 96.9%   | 96.6%   | 95.3%   |
> > > | Data Flow | TNR    | 17.8%   | 23.3%   | 26.7%   |
> > >
> > > t=0.5 short-circuits more aggressively, over-trusting the structural algorithm and producing very poor **TNR (19.9%, 17.8%)—many inequivalent pairs receive a premature "equivalent" verdict**. t=0.9 is harder to satisfy, triggering more LLM invocations and increasing cost, yet yields only marginal additional **TNR gains (+8.7%, +3.4%)** over t=0.7 while TPR continues to drop. By contrast, moving from t=0.5 to t=0.7 delivers the largest correctness improvement: **TNR on Control Flow jumps +14.4%** at a cost of only **-2.6% TPR**, and **TNR on Data Flow gains +5.5% at a negligible -0.3% TPR cost**. t=0.7 is therefore the point of inflection—capturing the large TNR improvement over t=0.5 without the diminishing returns and added LLM cost of t=0.9.
> > >
> > > As noted in our original rebuttal, **the threshold is a cost-optimization lever rather than a correctness lever—users can adjust it based on their own risk tolerance and budget, with a lower t favoring cheaper structural short-circuiting and a higher t favoring more thorough LLM-based analysis**. Our ablation across t $\in$ $(0.5, 0.7, 0.9)$ provides concrete TPR/TNR results to guide that decision.
> > >
> > > **Q2. Dispute Samples Ablation**
> > >
> > > We thank the reviewer for this suggestion. To address this concern, we ran the standalone baseline agent on all 416 dispute cases between MatchFixAgent and existing tools **(48.32% agreement with MatchFixAgent, 41.11% with tool, and the remaining with no yes/no verdict)**. However, as mentioned in our original rebuttal, these numbers do not convey any important meaning without ground-truths.
> > >
> > > To address this issue, we considered the **145** cases where manual investigation established ground truth (Section D.1.1), and we can directly measure baseline agent accuracy. On the 145 manually resolved cases, the baseline agent achieves only **47.73%** accuracy on the 88 cases where MatchFixAgent is correct—the cases that represent MatchFixAgent's unique advantage over existing tools. By contrast, the baseline achieves **56.14%** on the 57 cases where the existing tool is correct, confirming that the baseline agent systematically fails on precisely the hard, ambiguous translations that MatchFixAgent’s semantic analyzer and test generation are designed to handle.
> > >
> > > The **47.73%** accuracy on difficult, disputed cases (where our original ablation showed only a **42.3%** drop on easier, non-disputed cases) confirms that MatchFixAgent's multi-agent architecture is essential, not just helpful. The baseline agent's near-random performance on these difficult cases demonstrates that the semantic analyzer and test generation components are the core drivers of MatchFixAgent's accuracy advantage in resolving both straightforward and challenging translations.
> > >
> > > **Q3. Optimization**
> > >
> > > There are many directions that can be explored for optimizing both dollar cost and runtime of MatchFixAgent, which we outline below, but we leave this as future work. Please note though that MatchFixAgent’s current dollar cost and runtime compare favorably to the development cost of existing repository-level validation techniques, or even when compared to a salaried human software engineer manually performing validation. Compared to these two baselines, 309s and 1.22USD per translation pair is not “daunting”. A fully-automated translation tool that runs for a week and costs a few $100’s is still extremely useful in practice if the translation produced is high quality. Below are potential optimizations:
> > >
> > > - Multiple instances of MatchFixAgent can be run concurrently on functions that are not dependent on each either. We can determine function dependencies by constructing a call graph, which is a common program analysis technique. This approach would reduce total runtime, and does not require any change to the design of MatchFixAgent.
> > > - Use faster, cheaper LLMs or skip analysis for "trivial" translation pairs (e.g., simple getter/setter functions or those using only primitive types). Heuristics could detect these to avoid expensive semantic analysis or use a cheaper LLM.
> > > - A single instance of MatchFixAgent can process multiple related functions at once. Part of the overhead of running a coding agent is when the agent “orients” itself by reading code files to understand the codebase. By processing multiple related functions at once, we eliminate redundant "orientation" phases, reducing both runtime and cost.

---

### Official Review · Reviewer_u28r · 2026-03-10

**Soundness:** 3
**Presentation:** 2
**Significance:** 3
**Originality:** 2
**Overall Recommendation:** 4
**Confidence:** 3

**Summary:**

This paper studies the problem of evaluating the functional equivalence of translated code and repairing translation errors when they occur. The authors identify several limitations in existing approaches, including strong language specificity, high engineering overhead when adapting methods to new languages, and heavy reliance on existing test suites, which can lead to false equivalence claims and ineffective repair.
To address these issues, the paper proposes MatchFixAgent, a multi-agent architecture designed to assess equivalence and repair incorrect translations.
The system consists of three main components: a semantic analyzer, a test generation and repair agent, and a verdict agent.
The semantic analyzer extracts detailed program semantics, including control flow, data flow, I/O behavior, library usage, exception handling, and specifications.
Based on this analysis, the test generation and repair agent evaluates the equivalence between the source and translated programs, generates test cases in both the source and target languages, and optionally produces a patch to repair translations deemed inequivalent.
Finally, the verdict agent validates the conclusions of the test generation and repair agent and summarizes the results for end users.
The authors compare their approach against four repository-level code translation techniques. Their results show that MatchFixAgent more reliably produces correct equivalence or inequivalence judgments and significantly improves the repair of incorrect translations, achieving a 50.6% repair rate compared to 18.5% for the baselines.

**Compliance With Llm Reviewing Policy:**

Affirmed.

**Final Justification:**

This paper can be accepted provided the human study is conducted properly, as discussed during the rebuttal phase.

**Key Questions For Authors:**

1. **Ground-truth validation of equivalence judgments.**
   The evaluation relies primarily on comparisons with other systems. Could the authors annotate a small subset of cases (e.g., 50–100 translations) with independent human judgments of equivalence/inequivalence and report MatchFixAgent’s accuracy against this ground truth? This would help determine whether agreement cases are also correct and would substantially increase confidence in the reported results.

2. **Language extensibility effort.**
   The system requires Tree-sitter queries for CFG/DFG extraction to support new languages. Could the authors demonstrate the effort required to add one additional language (e.g., report implementation time and size of the query rules)? A concrete example would help substantiate the claim of language generality.

**Limitations:**

yes

**Strengths And Weaknesses:**

### Strengths:

* **More reliable equivalence judgments.** MatchFixAgent reaches an equivalence verdict more often than the compared techniques while agreeing with them in most cases. In instances where there is disagreement, MatchFixAgent is correct 60.7% of the time, suggesting that it provides more reliable judgments.

* **Broad language support.** MatchFixAgent supports the widest range of programming languages among the evaluated approaches. The framework also appears reasonably extensible to additional languages, for example by adding the necessary parsing support (e.g., via Tree-sitter).

* **Ablation analysis.** The authors conduct ablations using different coding agents and LLMs to evaluate the effectiveness of the proposed multi-agent framework, helping isolate the contribution of the architecture from the underlying models.

* **Well-designed semantic analysis component.** The semantic analyzer is carefully designed and extracts a broad set of program properties. The engineering effort behind this component could be valuable beyond the scope of this paper, for example in tools for code translation validation, code migration review, or potentially even general code review tasks with minor adaptations.

### Weaknesses:

* **Lack of ground-truth evaluation.** The most significant weakness is that the evaluation is not grounded in human-verified judgments of translation equivalence (as acknowledged in Appendix F: Threats to Validity). The paper argues that MatchFixAgent is more accurate than existing systems, which is supported by the analysis of cases where their verdicts disagree. However, for cases where all systems agree, it remains unclear whether they might be making the same mistake. This relative evaluation setup makes it difficult to establish objective correctness and also complicates comparisons with future techniques. Additionally, while there is no suggestion of misconduct, the fact that the disagreement analysis supporting the paper’s conclusions was conducted by the authors themselves introduces a potential conflict of interest. Incorporating a set of independently verified evaluations, or having disagreements reviewed by neutral third parties, would strengthen confidence in the results.

* **Dependence on the testing and repair agent.** The proposed multi-agent architecture appears to rely heavily on the testing and repair agent, which performs most of the core reasoning and output generation. Although the verdict agent can intervene and correct judgments, much of the system’s behavior seems driven by the capabilities of this component. The results also show noticeable performance variation between the OpenAI configuration (o4-mini-2025-04-16 + Codex) and the Anthropic configuration (Claude Sonnet 3.7 + Claude Code), with disagreement occurring in roughly one quarter of the cases. As discussed in Section 3.3.2, Codex tends to explore repositories less thoroughly and arrives at decisions more quickly. The framework could potentially be strengthened by introducing architectural components that explicitly encourage or enforce repository exploration, regardless of the underlying coding agent.

* **Lack of standardized tooling for agents.** Relatedly, the system does not provide a uniform set of tools for the testing and repair agent. As a result, integrating a new LLM into the framework may require implementing custom tooling, which increases the engineering overhead and reduces reproducibility.

* **Limited language-agnosticity in practice.** The paper’s claim of being language-agnostic is somewhat misleading. Supporting a new programming language requires implementing Tree-sitter queries for data flow and control flow graph analysis. The authors should either qualify this claim or demonstrate that this step can be reliably automated with coding agents. This may be particularly challenging for low-resource programming languages.

* **Presentation of results (Section 3).** The presentation of the main results could be improved. Tables 1 and 2 currently contain a large amount of information, which makes them difficult to parse. Focusing the main tables on agreement and disagreement rates, while moving the detailed breakdown of verdict types (EQ, NEQ, VF) to the appendix, would improve clarity. Similarly, the per-project breakdown may be unnecessarily detailed for the main paper and could also be moved to the appendix.

* **Terminology ambiguity.** The paper overloads the term “tools” to refer both to code translation systems and to agentic tools used by LLM-based agents. Using distinct terminology would reduce ambiguity and make it clearer when a component is purely symbolic, neural, or neuro-symbolic (e.g., an LLM agent equipped with tools).

---

> ### Author Rebuttal · Authors · 2026-03-31
>
> We would like to thank the reviewer for providing constructive feedback. Please find our response to your questions below. Please feel free to ask additional clarifying questions.
>
> **Q1. Ground-truth validation of equivalence judgments.**
>
> We appreciate the reviewer's point. Our human study on dispute cases aimed to show MatchFixAgent's superior accuracy over existing systems, not its absolute accuracy.
>
> Regardless, we performed a partial human study on non-dispute (equivalent/non-equivalent) cases within the short 6-day rebuttal period, conducted by one author due to time constraints. The second author commits to performing the study if **Reviewers u28r** and **f59W** are satisfied (Reviewer F59W Q2). The results and objective comments from the first author's study are uploaded to our public repo: https://doi.org/10.5281/zenodo.19343195 (Sheet rq1-eq-human-study-p2 in results.xlsx). We understand **Reviewer u28r**'s bias concern regarding author-investigators; therefore, we provide our sample-specific comments online for scrutiny.
>
> From the original experiments using Claude Sonnet 3.7, we uniformly sampled at most 5 equivalence judgements across 24 benchmarks and 4 validation systems, resulting in 110 source-target pairs. 88 were correctly validated as equivalent, and 22 were correctly validated as non-equivalent by both existing tools and MatchFixAgent. The human study found that 104/110 (94.5%) judgements were correct, meaning 6/110 (5.5%) were incorrect validations, indicating a need for more rigorous testing and oversight in MatchFixAgent. Representative incorrect examples follow.
>
> **Tool AND MatchFixAgent=Equivalent BUT human=otherwise:** see example1 source and target in: https://tinyurl.com/35vyup4j
>
> Explanation: The translated JavaScript incorrectly adds if (hue < 0) hue += 1.0; to handle negative hue values. Unlike Python's % operator, which ensures a non-negative result, the % operator in JavaScript (and other languages) can return negative values for negative operands. This divergence means the target code is not functionally equivalent to the Python source for negative hue inputs. Skel and MatchFixAgent failed to detect this bug due to insufficient testing.
>
> **Tool AND MatchFixAgent=Non-equivalent BUT human=otherwise:** see example2 source and target in: https://tinyurl.com/35vyup4j
>
> Explanation: Both functions compute the current time in slots since the Unix epoch by dividing the current Unix timestamp (in seconds) by (60 * TIME_SLOT_MINUTES). The logic is identical, with only language-specific differences in how current time is obtained and types used (unsign32 vs usize). RustRepoTrans and MatchFixAgent says these are functionally inequivalent because there is a compilation bug somewhere else in the codebase.
>
> **Q2. Language extensibility effort.**
>
> Our static analyzer module achieves PL-agnosticism without relying on complex Tree-Sitter queries to build CFG or DFG. Instead, it builds the CFG/DFG by directly traversing the parse tree generated by Tree-Sitter. To support a new PL, a user only needs to implement two files: model.py and cfg_builder.py.
>
> - model.py: This file defines all branching statements (e.g., if, elif, while, for for Python) specific to the new PL.
> - cfg_builder.py: The algorithm constructs the CFG and DFG simultaneously by incrementally traversing the parse tree. It combines non-branching statements into a single CFG node and adds edges based on branching statements' semantics while tracking data flow (e.g., parameters, local variables) for the DFG.
>
> To show low-effort extensibility, a coding agent (Claude Opus 4.5) implemented model.py and cfg_builder.py for Julia, a low-resource PL. By providing the existing implementations for six PLs, we prompted the agent to add Julia. In a single 5-minute, $2 process, the agent generated 442 lines of correct, manually verified code. The Julia implementation is publicly available: https://tinyurl.com/yukdefvj. A sample example with Julia code and generated output is also publicly available: https://tinyurl.com/p8h3psnv
>
> **===== Additional Concerns =====**
>
> **C1. Dependence on the testing and repair agent**
>
> We acknowledge the concern regarding the Test Generator and Repair Agent's importance. Its core function is to systematically generate and execute tests in both source and target PLs to validate the Semantic Analyzer's analysis. This validation is vital as it mitigates the risk of LLM hallucination (incorrect analyses) by enforcing concrete execution of the LLM's judgment. We agree that clearer instructions on codebase exploration will help resolve disagreements between Claude Code and Codex.
>
> **C2. Lack of standardized tooling for agents**
>
> The agentic coding tools (Read, Write, Edit) are largely LLM-independent. We confirmed this by successfully swapping Claude Sonnet 3.7 with Qwen3-Next-80B-A3B without modifying the tools (see **Reviewer d9RK, Q1**). The only requirement for the LLM is standard function calling support.

---

> > ### Author Rebuttal · Reviewer_u28r · 2026-04-04
> >
> > I thank the reviewers for addressing my concerns regarding extensibility and tool-calling, which I found satisfactory. However, I still have reservations about the core evaluation, particularly the human study.
> >
> > I briefly reviewed Sheet rq1-eq-human-study-p2 in results.xlsx, but I am not able to dedicate sufficient time to meaningfully serve as an additional annotator. More broadly, for the evaluation to be fully convincing, it would be important to include a more formal human study with multiple independent annotators and reported inter-annotator agreement.
> >
> > I understand that such additions may not be feasible within the rebuttal timeline, but I believe they are important for ensuring the overall soundness of the evaluation.

---

> > > ### Author Response · Authors · 2026-04-04
> > >
> > > We agree with the reviewer that additional annotators (with inter-annotator agreement score) will make our human study stronger and justify the soundness of MatchFixAgent. As **mentioned in our original rebuttal text**, we **commit to adding another annotator to our human study subject to the satisfaction of Reviewer u28r and f59W**. We are happy to see that our partial human study has been satisfactory and we will now plan to finish this on time for the final draft of our paper.
> > >
> > > We thank **Reviewer u28r** for recognizing that adding another annotator is not feasible during the rebuttal timeline. We are also happy to see your concerns about extensibility and tool-calling have been resolved.
> > >
> > > We believe there are no questions in the Rebuttal Acknowledgements, except for advice on adding another annotator which we plan to do. **In the light of this, we would be glad if Reviewer u28r could reconsider their original score** as their concerns have been addressed. We would love to address any further concerns you might have.

---

### Official Review · Reviewer_mvzk · 2026-03-11

**Soundness:** 3
**Presentation:** 3
**Significance:** 4
**Originality:** 3
**Overall Recommendation:** 4
**Confidence:** 4

**Summary:**

The authors focused on developing MATCHFIXAGENT, a language-agnostic multi-agent framework for validating whether repository-level code translations are functionally equivalent and for repairing incorrect translations. The framework combines semantic analysis, test generation, and repair to support multiple programming languages.

**Compliance With Llm Reviewing Policy:**

Affirmed.

**Key Questions For Authors:**

I was wondering if we can use this model for repairing mid-level languages as well.

**Limitations:**

- Is this code translation only applied to “high-level” languages?
- please explain more about the motivation for selecting Claude 3.7 Sonnet.

**Strengths And Weaknesses:**

Strengths

- The research change is interesting (validating and repairing codes).
- Proposes a clear multi-agent architecture (Semantic Analyzer, Test Generator & Repair Agent, Verdict Agent).=
- Evaluated on a large benchmark (2,219 translation pairs from 24 real-world projects across 6 language pairs).


Weaknesses
- The time complexity of the multi-agent model could be discussed!
- The discussion of limitations and failure cases could be more detailed.

---

> ### Author Rebuttal · Authors · 2026-03-31
>
> We would like to thank the reviewer for providing constructive feedback. Please find our response to your questions below. Please feel free to ask additional clarifying questions.
>
> **Q1. Is this code translation only applied to “high-level” languages? I was wondering if we can use this model for repairing mid-level languages as well.**
>
> Our technique extends beyond high-level languages (e.g., Python, JavaScript, Rust) and is applicable to mid-level languages as well. Assuming C is the mid-level language you are referring to, we note it was not a target Programming Language (PL) in our evaluation due to the lack of available code translation benchmarks that specifically translate to C. However, we hypothesize that our technique will perform effectively on any language that possesses sufficient representation within the LLM's pretraining data.
>
> **Q2. Please explain more about the motivation for selecting Claude 3.7 Sonnet.**
>
> We decided to choose Claude 3.7 as it was the best performing LLM available to us when developing MatchFixAgent. Our technique is LLM-agnostic as supported by our new set of experiments using Qwen3-Next-80B-A3B. Although it takes time and resources, we can also reproduce with Claude 4.5 if our paper is accepted, however, we do not think conclusions will significantly change.
>
> **===== Additional Concerns =====**
>
> **C1. The time complexity of the multi-agent model could be discussed.**
>
> Averaged across all six metric categories, Qwen3 is dramatically cheaper than Claude 3.7 on every tool tested. Alphatrans (126x cheaper), rustrepotrans (81x cheaper), oxidizer (25x cheaper) and skel (19x cheaper).
>
> | Model | Tool | Metric Category | Avg Input Token | Avg Output Token | Avg Runtime (ms) | Avg Cost (USD) |
> | :--- | :--- | :--- | :--- | :--- | :--- | :--- |
> | **Claude 3.7** | **alphatrans** | Control Flow | 437.0 | 86.9 | 2398.3 | 0.00261 |
> | | | Data Flow | 1124.5 | 122.8 | 3497.4 | 0.00522 |
> | | | IO | 782.4 | 205.0 | 5915.5 | 0.00542 |
> | | | Lib Equivalence | 700.4 | 235.9 | 6585.3 | 0.00564 |
> | | | Exception Error | 726.4 | 152.0 | 4537.2 | 0.00446 |
> | | | Spec | 726.4 | 311.1 | 7871.7 | 0.00685 |
> | | **oxidizer** | Control Flow | 1426.4 | 230.8 | 6218.9 | 0.00774 |
> | | | Data Flow | 2298.7 | 206.6 | 5593.1 | 0.00999 |
> | | | IO | 993.8 | 238.7 | 6594.3 | 0.00656 |
> | | | Lib Equivalence | 912.8 | 265.7 | 6861.8 | 0.00672 |
> | | | Exception Error | 937.8 | 248.3 | 6572.8 | 0.00654 |
> | | | Spec | 937.8 | 399.7 | 9853.6 | 0.00881 |
> | | **rustrepotrans**| Control Flow | 2247.7 | 266.4 | 7477.0 | 0.01074 |
> | | | Data Flow | 5083.8 | 275.7 | 8048.5 | 0.01939 |
> | | | IO | 1167.0 | 279.0 | 7782.3 | 0.00769 |
> | | | Lib Equivalence | 1086.0 | 318.4 | 8965.1 | 0.00803 |
> | | | Exception Error | 1111.0 | 287.8 | 7617.4 | 0.00765 |
> | | | Spec | 1111.0 | 491.5 | 12353.4 | 0.01071 |
> | | **skel** | Control Flow | 1163.1 | 201.4 | 5876.9 | 0.00651 |
> | | | Data Flow | 3621.2 | 258.6 | 7575.4 | 0.01474 |
> | | | IO | 907.7 | 218.5 | 6663.9 | 0.00600 |
> | | | Lib Equivalence | 826.7 | 233.5 | 6711.2 | 0.00598 |
> | | | Exception Error | 851.7 | 157.5 | 4957.9 | 0.00492 |
> | | | Spec | 851.7 | 368.4 | 9496.9 | 0.00808 |
> | --- | --- | --- | --- | --- | --- | --- |
> | **Qwen3** | **alphatrans** | Control Flow | 65.9 | 13.3 | 304.2 | 0.00002 |
> | | | Data Flow | 191.1 | 18.6 | 598.9 | 0.00003 |
> | | | IO | 131.0 | 42.9 | 990.7 | 0.00005 |
> | | | Lib Equivalence | 118.3 | 43.5 | 952.1 | 0.00004 |
> | | | Exception Error | 121.5 | 36.9 | 854.3 | 0.00004 |
> | | | Spec | 121.5 | 65.1 | 1399.9 | 0.00006 |
> | | **oxidizer** | Control Flow | 1186.9 | 168.5 | 3957.4 | 0.00024 |
> | | | Data Flow | 1928.8 | 159.1 | 3726.5 | 0.00030 |
> | | | IO | 833.6 | 297.3 | 6532.9 | 0.00031 |
> | | | Lib Equivalence | 770.6 | 250.3 | 5640.9 | 0.00026 |
> | | | Exception Error | 786.6 | 241.8 | 6100.1 | 0.00026 |
> | | | Spec | 786.6 | 408.7 | 10097.8 | 0.00039 |
> | | **rustrepotrans**| Control Flow | 639.2 | 75.0 | 2242.2 | 0.00012 |
> | | | Data Flow | 2125.5 | 57.9 | 1577.4 | 0.00024 |
> | | | IO | 303.8 | 96.6 | 2607.4 | 0.00010 |
> | | | Lib Equivalence | 284.0 | 84.7 | 2123.0 | 0.00009 |
> | | | Exception Error | 289.0 | 91.1 | 3153.7 | 0.00010 |
> | | | Spec | 286.8 | 140.6 | 3104.4 | 0.00014 |
> | | **skel** | Control Flow | 949.7 | 153.5 | 3640.0 | 0.00021 |
> | | | Data Flow | 4951.7 | 195.3 | 4867.5 | 0.00060 |
> | | | IO | 761.1 | 245.6 | 5834.6 | 0.00026 |
> | | | Lib Equivalence | 698.1 | 221.2 | 5237.1 | 0.00024 |
> | | | Exception Error | 714.1 | 193.7 | 4896.8 | 0.00021 |
> | | | Spec | 714.1 | 378.8 | 8539.1 | 0.00036 |
>
> **C2. The discussion of limitations and failure cases could be more detailed.**
>
> We include discussion on limitations of MatchFixAgent and competing techniques in our submission (Appendix D). Please refer to our anonymous artifacts (https://doi.org/10.5281/zenodo.19343195) where we provide an excel sheet (results.xlsx) for more investigation comments for every sample of human study.

---

### Official Review · Reviewer_d9RK · 2026-03-13

**Soundness:** 4
**Presentation:** 4
**Significance:** 3
**Originality:** 3
**Overall Recommendation:** 5
**Confidence:** 4

**Summary:**

The paper proposes an LLM-based, programming-language-agnostic framework that automatically validates whether repository-level code translations are functionally equivalent and repairs incorrect translations when they are not. This paper argues that prior validation/repair systems suffer from three main problems: they are hard to generalize across many programming-language pairs because they require substantial language-pair-specific engineering; they depend on incomplete test suites or fuzzing, which can cause false equivalence or false inequivalence; and their repair strategies are weak, often relying on manual fixing or shallow reprompting. MATCHFIXAGENT addresses these issues by replacing pair-specific validation logic with a PL-agnostic, multi-agent LLM workflow supported by lightweight static analyses and in-the-loop test generation plus repair. The overview architecture has three main parts: a Semantic Analyzer, a Test Generator & Repair Agent, and a Verdict Agent. The Semantic Analyzer first studies the source function and translated function through six parallel semantic views: control flow, data flow, input/output behavior, library API equivalence, exception/error behavior, and inferred or explicit specifications; some of the structural analyses use lightweight similarity thresholds to skip unnecessary LLM calls and reduce cost. The evaluation uses benchmarks collected from prior repository-level translation works as OXIDIZER, ALPHATRANS, SKEL, and RUSTREPOTRANS. The authors compare MATCHFIXAGENT against those prior systems on validation and repair, mainly using Claude 3.7 Sonnet with Claude Code, while also testing adaptability with o4-mini plus Codex on a sample. Overall, MATCHFIXAGENT returns verdicts for 99.2% of all pairs.

**Compliance With Llm Reviewing Policy:**

Affirmed.

**Final Justification:**

Authors provided rigorous analysis in rebuttal phase. Importantly,author provided the experiment with open LLM and shows that their approach can be applied with open LLM without decreasing the accuracy significantly.

**Key Questions For Authors:**

1. Can we replace closed LLM by fine-tuned open LLMs (Qwen, Mistral models) for the Semantic Analyzer? Will that configuration affect the overall accuracy?
2. Ablation study on the budget of using close LLMs for experiment.
2. How does the verdict agent address the correctness of translated code in terms of readability besides functional equivalent?
3. How can MatchFixAgent ensure that the test generation by Test Generator and Repair Agent is correct?
4. A study on analyzing the cases when LLM perform good/ bad with the Semantic Analyzer is recommended.

**Limitations:**

The main concern of the reviewer is the risk of the correctness of LLMs in multiple modules of MatchFixAgent. Approaches for judging the qualities of LLMs in these modules is needed.

**Strengths And Weaknesses:**

Strength:
- The paper is well-written.
- The idea of a rigorous checking program equivalence by a semantic analyzer is a good direction for improving code translation.
- Rigorous experiments with valid baselines.
- Valid selection of LLM Agent.
- This approach can be generalized for multiple languages.

Weakness:
- The Semantic Analyzer, Test Generator & Repair Agent, and Verdict Agent integrated multiple LLM query-answering. It will cause the risk to the correctness of LLMs, especially for recently appeared programming languages. A technique of judging how well the LLM performed in control-flow and data-flow generation is needed.
- Existing work [1] shows that each LLM has advantages in each aspect. Instead of using one LLM for all steps of the Semantic Analyzer, authors are recommended to use various LLMs to see which LLMs perform best for each module.
- The verdict agent only measured the functionality equivalence and didn't consider other aspects such as readability or naming convention.
- The framework requires multiple phases of LLM interaction over close models. An ablation study over the average budgets is recommended.
- Lack of using fine-tuning or other LLM optimization approaches for LLMs in the Semantic Analyzer.


References:
1. CodeUltraFeedback: An LLM-as-a-Judge Dataset for Aligning Large Language Models to Coding Preferences
Martin Weyssow, Aton Kamanda, Xin Zhou, Houari Sahraoui

---

> ### Author Rebuttal · Authors · 2026-03-31
>
> We would like to thank the reviewer for providing constructive feedback. Please find our response to your questions below. Please feel free to ask additional clarifying questions.
>
> **Q1. Can we replace closed LLM by fine-tuned open LLMs (Qwen, Mistral models) for the Semantic Analyzer? Will that configuration affect the overall accuracy?**
>
> MatchFixAgent is LLM-agnostic and easily adapts to new models. To show its open-source adaptability, we are reproducing Table 1 with the state-of-the-art Qwen3-Next-80B-A3B (released Feb 2026). Experiments are ongoing due to long runtimes (completed projects include Skel, Oxidizer, AlphaTrans: commons-cli; RustRepoTrans: deltachat-core). Current results show MatchFixAgent (Qwen3) agreement with existing tools is 74.2%, nearly identical to Claude 3.7's 74.4%, suggesting similarity between the open-source and closed LLMs.
>
> |    **Tool**   | **Total # Trans. Pairs** |  **Tool (EQ)** | **Tool (NEQ)** | **Tool (VF)** | **MatchFixAgent (EQ)** | **MatchFixAgent (NEQ)** | **MatchFixAgent (VF)** |  **Agreement** |
> |:-------------:|:------------------------:|:--------------:|:--------------:|:-------------:|:----------------------:|:-----------------------:|:----------------------:|:--------------:|
> |    Oxidizer   |            192           |   138 (71.9)   |    53 (27.6)   |    1 (0.5)    |       128 (66.7)       |        63 (32.8)        |         1 (0.5)        |   117 (61.6)   |
> |   AlphaTrans  |            273           |   210 (76.9)   |    24 (8.8)    |   39 (14.3)   |       214 (78.4)       |        53 (19.4)        |         6 (2.2)        |   173 (75.2)   |
> |      Skel     |            337           |   276 (81.9)   |    19 (5.6)    |   42 (12.5)   |       250 (74.2)       |        73 (21.7)        |        14 (4.2)        |   229 (80.9)   |
> | RustRepoTrans |            125           |    54 (43.2)   |    69 (55.2)   |    2 (1.6)    |        39 (31.2)       |        84 (67.2)        |         2 (1.6)        |     92 (76)    |
> |   **Total**   |          **927**         | **678 (73.1)** | **165 (17.8)** |  **84 (9.1)** |     **631 (68.1)**     |      **273 (29.4)**     |      **23 (2.5)**      | **611 (74.2)** |
>
> **Q2. Ablation study on the budget of using close LLMs for experiment.**
>
> Please refer to our response to **Reviewer mvzk’s C1** for a fine-grained analysis of MatchFixAgent’s time and cost complexity for closed and open-source LLMs.
>
> **Q3. How does the verdict agent address the correctness of translated code in terms of readability besides functional equivalent?**
>
> In MatchFixAgent, our main priority is to address the functional correctness between source and target PL code snippets. Non-functional metrics, although essential, are not in our scope in this work. However, it's worth noting that a user of our tool can easily introduce non-functional metrics, e.g., readability into the MatchFixAgent prompts.
>
> **Q4. How can MatchFixAgent ensure that the test generation by Test Generator and Repair Agent is correct?**
>
> The Test Generator and Repair Agent systematically generates and executes tests in both source and target programming languages to validate the six analyses from the Semantic Analyzer. This validation is critical as LLMs can hallucinate. By enforcing concrete execution of the LLM's judgment, the agent—viewed as a "heuristic"—mitigates hallucination risk in assessing equivalence. While we don't claim the generated tests are inherently correct, our human study (Appendix, Section D.1.1) supports their correctness and confirms potential hallucinations.
>
> **Q5. A study on analyzing the cases when LLM perform good/ bad with the Semantic Analyzer is recommended.**
>
> We used a confusion matrix to compare each semantic analyzer's verdict against the final verdict and simplify the results with metrics. Our analysis shows that when using Claude, the Control Flow has a **31.2% FPR (Control Flow=Not Equivalent, Final Verdict=Equivalent)**, and the Data Flow has a **25.9% FNR (Data Flow=Equivalent, Final Verdict=Not Equivalent)**. While any analyzer can give misleading hints, the agent's execution ability helps detect incorrect analyses.
>
> | Semantic Analyzer      |  TP   |  FN |  TN |  FP | TPR   | FNR   | TNR   | FPR   |
> |------------------------|------:|----:|----:|----:|------:|------:|------:|------:|
> | Control Flow           | 1,404 | 438 | 229 | 104 | 76.2% | 23.8% | 68.8% | 31.2% |
> | Data Flow Path         | 1,459 | 510 | 155 |  52 | 74.1% | 25.9% | 74.9% | 25.1% |
> | Input/Output Mapping   | 1,437 | 321 | 347 |  71 | 81.7% | 18.3% | 83.0% | 17.0% |
> | Library API Equiv.     | 1,481 | 482 | 178 |  18 | 75.4% | 24.6% | 90.8% |  9.2% |
> | Exception/Error Hdl.   | 1,420 | 425 | 243 |  91 | 76.9% | 23.1% | 72.8% | 27.2% |
> | Specifications         | 1,406 | 308 | 359 |  93 | 82.0% | 18.0% | 79.4% | 20.6% |
> | **Average**            | **1,435** | **414** | **252** | **72** | **77.7%** | **22.3%** | **78.3%** | **21.7%** |

---

> > ### Author Rebuttal · Reviewer_d9RK · 2026-04-02
> >
> > The authors addressed my concerns in the rebuttal phase. So I adjust my score.

---

> > > ### Author Response · Authors · 2026-04-03
> > >
> > > Thank you for your thoughtful comments and we are glad to have addressed your concerns to improve our manuscript.

---

### Decision · Program_Chairs · 2026-04-30

**Decision:**

Accept (regular)

**Comment:**

We thank the authors for their submission. Reviewers generally agreed that this is a worthwhile topic and that the proposed agent is effective and should have a nontrivial impact on the overhead of supporting new languages. Reviewers did note a number of concerns, including the high cost per instance, the lack of ground-truth labels, and the heavy reliance on TreeSitter, which produces relatively simple program analysis. The authors responded to these issues in great detail, resulting in the majority of reviewers agreeing that the work is fit for acceptance once the authors integrate the proposed changes.